# MF-LLM: Simulating Population Decision Dynamics via a Mean-Field Large Language Model Framework

**Qirui Mi**[1,2,3]**, Mengyue Yang**[4]**, Xiangning Yu**[5]**, Zhiyu Zhao**[1,2]**, Cheng Deng**[6]**,**
**Bo An**[3]**, Haifeng Zhang**[1,2*]**, Xu Chen**[7*]**, Jun Wang**[8*]
[1]Institute of Automation, Chinese Academy of Sciences
[2]School of Artificial Intelligence, Chinese Academy of Sciences
[3]Nanyang Technological University    [4]University of Bristol
[5]Tianjin University    [6]Shanghai Jiao Tong University
[7]Renmin University of China    [8]University College London

## Abstract

Simulating collective decision-making involves more than aggregating individual behaviors; it emerges from dynamic interactions among individuals. While large language models (LLMs) offer strong potential for social simulation, achieving quantitative alignment with real-world data remains a key challenge. To bridge this gap, we propose the **M**ean-**F**ield **LLM** (**MF-LLM**) framework, the first to incorporate mean field theory into LLM-based social simulation. MF-LLM models bidirectional interactions between individuals and the population through an iterative process, generating population signals to guide individual decisions, which in turn update the signals. This interplay produces coherent trajectories of collective behavior. To improve alignment with real-world data, we introduce **IB-Tune**, a novel fine-tuning method inspired by the **I**nformation **B**ottleneck principle, which retains population signals most predictive of future actions while filtering redundant history. Evaluated on a real-world social dataset, MF-LLM reduces KL divergence to human population distributions by **47%** compared to non-mean-field baselines, enabling accurate trend forecasting and effective intervention planning. Generalizing across 7 domains and 4 LLM backbones, MF-LLM provides a scalable, high-fidelity foundation for social simulation.

## 1 Introduction

Simulating how population-level decisions evolve over time is essential for predicting the spread of public opinion [3], household responses to policy shocks [17], and crowd dynamics during emergencies [15]. Unlike static aggregation of individual behaviors, collective decision-making emerges from dynamic interactions among individuals [33, 34, 5], where each agent's choices are shaped both by private observations and by the evolving actions of others [7]. These individual decisions, in turn, continuously reshape the population distribution, creating a feedback loop that drives population dynamics. Classic agent-based models simulate this loop using handcrafted rules [6, 25], but often lack realism and fail to generalize beyond narrow scenarios.

Recent studies show that large language models (LLMs) can endow agents with rich world knowledge and generative capabilities, enabling simulations ranging from "generative towns" [31, 48] to social-media environments [43, 45, 35], election forecasting [44], and macroeconomic modeling [22]. State-of-the-art simulations such as OASIS [43] and AgentSociety [32] excel at environment modeling but remain *prompt-centric* in agent decision-making—relying on manually crafted role templates, heuristic memory-retrieval rules, and fixed interaction schedules. These expert-crafted heuristics

---

[*]Corresponding author: jun.wang@ucl.ac.uk, xu.chen@ruc.edu.cn, haifeng.zhang@ia.ac.cn

39th Conference on Neural Information Processing Systems (NeurIPS 2025).

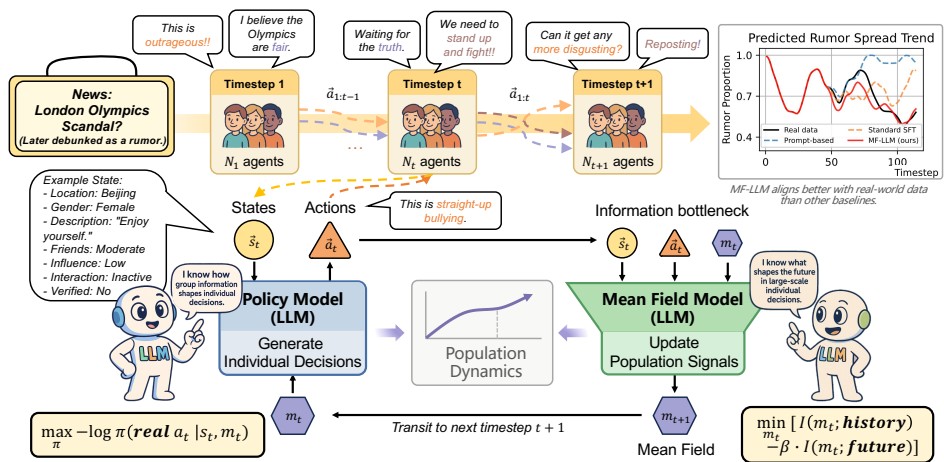

Figure 1: **MF-LLM framework for simulating population decision dynamics.** When an event occurs (e.g., a rumor), individuals make sequential decisions (e.g., "*This is outrageous!*") influenced by evolving collective behavior such as public opinion. Early decisions shape collective behavior, which in turn influences future actions, creating a feedback loop. MF-LLM captures this loop by alternating two LLM-based modules: a *policy model* that generates individual decisions based on personal states and population signals, and a *mean field model* that updates population signals from new actions. This iterative process closely aligns with real-world population dynamics (top right).

enhance *qualitative plausibility* but lack principled alignment to real-world behavioral data. This limits their applications in tasks requiring *quantitative fidelity* to real-world behavior, such as ex-ante policy evaluation or counterfactual intervention planning. **Bridging this realism gap—transforming *qualitative* simulations into social simulations that *quantitatively* match real data—remains an open challenge** [38].

Despite recent advances, three key challenges hinder progress toward social simulation that aligns with real-world data. **(C1) Prompt-based heuristics lack data alignment.** Many existing simulations rely on carefully crafted prompts or scripted interaction schedules. While these may produce plausible narratives, they fail to track the evolving dynamics of real-world populations. As shown in Fig. 1 (top right), the prompt-based simulation (blue) initially follows the real-world rumor trend, but quickly diverges as collective behavior shifts. **(C2) Supervised fine-tuning overlooks interaction dynamics.** Supervised fine-tuning (SFT) offers a direct path to align LLM agents with real-world data. However, it treats each decision as independent, ignoring how agents influence one another over time. As a result, SFT may fit individual behavior but fails to reproduce collective dynamics [28, 7]. In Fig. 1 (top right), SFT-based simulation (orange) fails to follow the real-world rumor trajectory. **(C3) Balancing fidelity and scalability remains a core challenge.** As agent population grows, simulating agents' interactions becomes increasingly complex and costly. Yet such interactions are essential for reproducing real-world dynamics, as collective behavior emerges from agent-to-agent influence. Balancing fidelity to real-world behavior with scalability thus remains a core challenge.

**Contributions**. To address the challenges of quantitative population simulation (C1–C3), we make the following contributions:

1. **MF-LLM: scalable simulation via mean-field agent-population interaction.** We introduce *Mean-Field LLM* (MF-LLM; see Fig. 1), a framework for simulating population dynamics. Inspired by **mean-field theory** [19], MF-LLM replaces agent-to-agent interactions with agent–population interactions. This avoids the combinatorial cost of modeling all agent-to-agent interactions (**C3**) while preserving key feedback dynamics (**C2**). To implement this, MF-LLM alternates between two LLM-based modules to generate the trajectory of collective decisions over time: **(i)** a *mean-field model* that generates dynamically updated population signals based on past individual actions, and **(ii)** a *policy model* that generates subsequent individual actions informed by these population signals. This design reflects the bidirectional interaction between individual decisions and population behavior (**C2**), while remaining scalable to large populations (**C3**).

2. **IB-Tune: a data-driven algorithm for real-data alignment.** To overcome the limitations of SFT (**C1**), we propose a fine-tuning algorithm, *IB-Tune*, to optimize the above LLM modules. **(i)** The mean-field model is trained with an **information bottleneck** objective, extracting population signals that carry predictive information about future actions while discarding redundant history.

**(ii)** The policy model is trained via negative log-likelihood supervision against real-world actions. This creates a feedback loop: accurate population signals lead to more human-like actions, which in turn improve the learning of decision-relevant population signals (**C2**).

3. **Evaluation and key findings.** We evaluate MF-LLM on the WEIBO corpus ($\sim 4{,}500$ real-world events): **(i)** MF-LLM aligns closely with real-world population trends (Fig. 2), reduces KL divergence by **47%** over non-mean-field baselines, outperforms baselines across semantic dimensions (Fig. 3), and generalizes across domains (Fig. 4) and LLM backbones (Table 1). **(ii)** Ablation results show that both the mean-field module and IB-Tune are critical, with up to 118% degradation when removed (Table 2). **(iii)** We further examine MF-LLM's real-world applicability—forecasting and intervention planning (Fig. 5)—and find that exogenous signal injection further improves fidelity (Fig. 8).

Code is available at `https://github.com/Miracle1207/Mean-Field-LLM`.

## 2 Related Work

A full discussion of related work appears in Appendix H; we highlight key directions here.

**Agent-Based Models (ABMs).** ABMs have long served as a foundation for modeling complex social systems through local agent interactions, with seminal applications in artificial societies [10], crowd behavior [15], markets [36], ecosystems [14], and public policy [2, 26]. While effective for exploring emergent phenomena, classical ABMs often rely on manually designed rules, limiting scalability and generalization to real-world complexity.

**LLMs for Social Simulation.** Recent work has explored using large language models (LLMs) to enhance social simulation, leveraging their generative capabilities and world knowledge. Systems such as OASIS [43, 45], ElectionSim [44], EconAgent [22, 42], and toolkits like GenSim [35] and AgentSociety [32] demonstrate promising use cases across domains. However, many current LLM-based agents operate on prompt-engineered templates, scripted roles, and heuristic memory, with limited alignment to real-world data [8, 23], hindering their quantitative reliability.

**Mean Field Theory for Scalable Interactions.** Mean field theory offers a principled way to model large-scale agent interactions by replacing explicit pairwise interactions with population-level dynamics [19, 16, 41]. It has been applied in social influence [4, 40], energy systems [9], traffic [11], and economics [27]. While prior work shows its success in reinforcement learning, extending it to LLMs is non-trivial: modeling populations where each agent is an LLM makes dynamic updates computationally expensive. These challenges motivate our MF-LLM framework.

## 3 Mean-Field LLM

We begin by formalizing the problem of simulating population decision dynamics (§3.1), then introduce **MF-LLM** framework (§3.2) and **IB-Tune** algorithm (§3.3) to align with real-world data.

### 3.1 Problem Statement

We aim to simulate population decision-making in text-driven environments, where *population dynamics* capture evolving collective behaviors emerging from decentralized decisions of interacting agents (**C2**). Consider a population of $N$ agents making sequential decisions in a shared, language-based space. At each timestep $t$, a subset of $N_t$ agents ($\sum_t N_t = N$) acts simultaneously based on textual states encoding personal attributes (e.g., roles, preferences) and environmental observations (e.g., task descriptions). We denote the agent states as $\vec{s}_t = (s_t^1, \ldots, s_t^{N_t})$, where $s_t^i \in \mathcal{S}$ represents agent $i$'s state. Each agent selects an action $a_t^i \in \mathcal{A}$, forming a joint action vector $\vec{a}_t = (a_t^1, \ldots, a_t^{N_t})$, which influences the environment and future behaviors. The parameter $N_t$ controls simulation granularity: larger $N_t$ enables efficient, coarse rollouts, while smaller $N_t$ captures finer dynamics.

Unlike traditional MDPs or Markov games with discrete actions, our language-based simulation operates in open-ended natural language, where both state space $\mathcal{S}$ and action space $\mathcal{A}$ are unbounded free-form text. We model agent decision-making with an LLM-driven policy, leveraging LLMs' generative power to produce context-aware, semantically rich behaviors grounded in broad priors.

## 3.2 Mean-Field LLM Framework

To quantitatively simulate population dynamics, we propose the *Mean-Field LLM (MF-LLM)* framework—the first to introduce mean-field theory [19] into LLM-driven social simulation. We now describe the core components of the MF-LLM framework.

**Scalable Interaction Modeling via Mean Field.** Simulating all agent-to-agent interactions becomes intractable as population size and time horizon grow. While full interaction histories can be encoded for small groups, this approach does not scale to realistic population settings **(C3)**. To address this, MF-LLM introduces a *mean-field approximation* [41, 27] to model agent–population dynamics without tracking all pairwise interactions. Unlike agent-centric memory designs that store and retrieve individual experiences [31, 13], the **mean field** in MF-LLM encodes a population-centric, continuously evolving signal that influences subsequent agent decision-making. It is initialized as an empty string $m_0$ and recursively updated by an LLM $\mu$, referred to as the *mean field model*:

$$m_t \leftarrow \mu(m_{t-1}, \vec{s}_{t-1}, \vec{a}_{t-1}), \tag{1}$$

where $(\vec{s}_{t-1}, \vec{a}_{t-1})$ represent the latest agent states and actions. Instead of passively accumulating interaction histories, MF-LLM dynamically distills decision-critical information from evolving population trajectories. This mechanism preserves essential social dynamics while maintaining scalability within LLM context limits. Detailed mean-field prompts are provided in Appendix F.4.

To stabilize early-stage simulation, we design a **warm-up phase** where the mean field is updated with real actions $\vec{a}_t^*$ from background context. After $t_{\text{warmup}}$, the simulation proceeds with policy-generated actions. $t_{\text{warmup}}$ is determined based on task-specific considerations, as discussed in the experiments.

**Decision-Making via Policy Model.** Given the current mean field $m_t$ and individual state $s_t^i$, each agent selects an action based on a language-model-driven policy. This policy maps textual inputs to a distribution over possible textual actions:

$$a_t^i \sim \pi^i(\cdot \mid s_t^i, m_t), \tag{2}$$

where $\pi^i : \mathcal{S} \times \mathcal{M} \to \mathcal{A}$ denotes the agent's LLM-based policy. By operating in unbounded language spaces, MF-LLM captures highly expressive, context-sensitive, and adaptive decision-making, bridging closer to real-world complexity. Detailed policy prompts are shown in Appendix F.4.

**Environment Transition Dynamics.** After all $N_t$ agents complete their decisions at timestep $t$, the system transitions to the next state. State transitions depend on the current joint state-action pair $(\vec{s}_t, \vec{a}_t)$ and the mean field $m_t$:

$$\vec{s}_{t+1} \sim P(\cdot \mid \vec{s}_t, \vec{a}_t, m_t), \tag{3}$$

where $P$ is the environment-specific stochastic transition function. This setup enables MF-LLM to capture both individual decision-making and population-level evolution over time. Pseudocode 1 summarizes the overall framework. We also provide intuitive examples in Appendix C to illustrate how MF-LLM models complex and heterogeneous interactions.

---

**Pseudocode 1** Simulation Procedure of the MF-LLM Framework
___

1: **Input:** Number of subset agents $N_t$, time horizon $T$, warm-up horizon $t_{\text{warmup}}$.
2: **Initialize:** Mean field $m_0$ as empty string, agent states $\{s_0^i\}_{i=1}^N$.
3: **for** $t = 0$ **to** $T - 1$ **do**
4:      **if** $t \leq t_{\text{warmup}}$ **then**                       ▷ Warm-up phase using background actions
5:          Retrieve real actions: $\vec{a}_t \leftarrow \vec{a}_t^*$
6:      **else**                                   ▷ Simulation phase using LLM policy
7:          **for each** agent $i$ **in** active set **do**          ▷ subset of $N_t$ agents active at time $t$
8:              Observe $s_t^i, m_t$; generate action $a_t^i \sim \pi^i(\cdot \mid s_t^i, m_t)$
9:              Form joint action: $\vec{a}_t \leftarrow (a_t^1, \ldots, a_t^{N_t})$
10:     Update mean field: $m_{t+1} \leftarrow \mu(m_t, \vec{s}_t, \vec{a}_t)$
11:     Environment transition: $\vec{s}_{t+1} \sim P(\cdot \mid \vec{s}_t, \vec{a}_t, m_t)$
12: **Output:** Sequence of simulated actions $\{\vec{a}_t\}_{t=0}^T$

---

### 3.3 IB-Tune Algorithm

While MF-LLM captures population dynamics using pretrained LLMs, its alignment with real-world data can be further improved through targeted fine-tuning. We propose *IB-Tune*, a fine-tuning algorithm that optimizes: (1) the mean field model, to extract **predictive** population-level signals; and (2) the policy model, to generate behaviorally **realistic** actions conditioned on these signals.

**Mean Field Model Optimization.** In the MF-LLM framework (Section 3.2), the mean field $m_t$ serves as a dynamic population signal that encodes predictive information about future individual actions. To enhance its predictive value and compress irrelevant history, we optimize $m_t$ via the **Information Bottleneck (IB)** principle [37, 1], which learns representations that retain only task-relevant information. Formally, given input variables $X$ and target variables $Y$, the IB objective seeks a latent variable $Z$ that minimizes the trade-off:

$$\min_Z I(Z; X) - \beta \cdot I(Z; Y), \tag{4}$$

where $I(\cdot; \cdot)$ is mutual information and $\beta$ balances compression and prediction.

Applying this principle, we construct the mean field $m_t$ as a *textual representation* that retains the essential information to predict individual actions in the next step $Y = \{a_t^{*i}\}_{i=1}^{N_t}$ while discarding irrelevant details from historical trajectories $X = \{(\vec{s}_\tau^*, \vec{a}_\tau^*)\}_{\tau=0}^{t}$. This leads to the IB-style objective:

$$\min_{m_t} I(m_t; X) - \beta \cdot I(m_t; Y), \tag{5}$$

To approximate the predictive term $I(m_t; Y)$, we leverage the intuition that an informative mean field should enable the policy to better reproduce real-world behavior. Following standard IB practice [1], we approximate it via the log-likelihood of observed actions under the policy model:

$$I(m_t; Y) \approx \sum_{i=1}^{N_t} \log \pi(a_t^{*i} \mid s_t^i, m_t). \tag{6}$$

For the compression term $I(m_t; X)$, we adopt a variational upper bound that makes the objective tractable. Full derivation of this bound is shown in Appendix G.1. Let $\mu_\phi(m_t \mid X)$ denote the learned posterior distribution and $r(m_t)$ a fixed prior induced by a pretrained LLM. Following standard IB approximations [1], we bound the term as:

$$I(m_t; X) \leq \mathbb{E}_{p(X)}\left[\text{KL}\big(\mu_\phi(m_t \mid X) \,\|\, r(m_t)\big)\right] = \mathbb{E}_{p(X)}\left[\mathbb{E}_{\mu_\phi(m_t \mid X)}\big[\log \mu_\phi(m_t \mid X) - \log r(m_t)\big]\right].$$

In practice, we instantiate $X$ as the immediate trajectory $\{m_{t-1}, \vec{s}_{t-1}, \vec{a}_{t-1}\}$ to balance predictive richness and computational efficiency. Combining the above, the final objective for optimizing the mean field model rewrites Eq. 5 as:

$$\mathcal{L}_{\text{mean-field}} = \sum_{t=1}^{T} \left[\mathbb{E}_{p(X_t)}\text{KL}\big(\mu_\phi(m_t \mid X_t) \,\|\, r(m_t)\big) - \beta \sum_{i=1}^{N_t} \log \pi\big(a_t^{*i} \mid s_t^i, m_t\big)\right], \tag{7}$$

where $m_t \sim \mu_\phi(\cdot \mid m_{t-1}, \vec{s}_{t-1}, \vec{a}_{t-1})$. This objective encourages $m_t$ to serve as a compact yet predictive population signal, improving the fidelity of the decision-making of the downstream agent.

**Policy Model Optimization.** The policy model $\pi$ governs individual decision-making conditioned on private states and the mean field. Each agent observes its own state $s_t^i$ and the shared mean field $m_t$, selecting an action $a_t^i$ accordingly. We assume a factorized policy structure, where the joint action distribution decomposes as:

$$\pi(\vec{a}_t \mid \vec{s}_t, m_t) = \prod_{i=1}^{N_t} \pi(a_t^i \mid s_t^i, m_t),$$

with all agents sharing the same policy model $\pi$. Agent-specific behavior arises naturally from personalized state inputs $s_t^i$, expressed in rich natural language. This design ensures scalability and generalization across heterogeneous populations.

To align the policy model with real-world agent behavior, we minimize the negative log-likelihood of observed actions:

$$\mathcal{L}_{\text{policy}} = -\sum_{t=1}^{T} \sum_{i=1}^{N_t} \log \pi(a_t^{*i} \mid s_t^i, m_t), \tag{8}$$

where $a_t^{*i}$ denotes the ground-truth action of agent $i$ at time $t$. This objective encourages the policy model to generate behaviorally realistic individual actions conditioned on both private states and evolving population signals, while continuously tracking the evolution of population dynamics.

# 4 Experiments

We evaluate MF-LLM on its ability to simulate population decision dynamics using the WEIBO corpus. Our experiments are structured around two core questions:

1. **How Real is MF-LLM? Evaluating Its Fidelity and Generalization** (§4.3). We assess MF-LLM's ability to match real-world decision trajectories across temporal trends, semantic dimensions, and diverse social domains, and evaluate its robustness across multiple base LLMs.

2. **What Drives MF-LLM? Dissecting the Mean Field and IB-Tune Mechanisms** (§4.4). We conduct ablations to assess the necessity of the mean-field module and the IB-Tune algorithm for ensuring simulation fidelity.

## 4.1 Evaluation Setup

**Dataset.** We conduct experiments on both the WEIBO corpus and a TWITTER dataset. The WEIBO corpus [24] contains over 5,000 real-world events, hundreds of temporally ordered user responses per event, and rich user profiles. Its scale, accessibility, and granularity make it well-suited for modeling population decision dynamics. Details are provided in Appendix F.1. The TWITTER dataset[2] contains a large collection of tweet–reply pairs centered on popular keywords, while a MARKET BEHAVIOR dataset constructed from Bitcoin-related tweet–reply interactions is used to simulate collective bullish/bearish sentiment dynamics. Results on the TWITTER and MARKET BEHAVIOR datasets show consistent trends with those on WEIBO; due to page limitations, these results are presented in Appendix B.2.

**Baselines.** Our baselines are drawn from existing LLM-based frameworks (e.g., AgentSociety[32], OASIS[43]) based on their designs for dynamic population interactions (Details in Appendix F.2).

- **State**[44]: LLM conditioned only on *user profile* and *event topic*.
- **Recent**[45, 32]: State + $k$ most recent actions from other agents (e.g., Top-$k$ comments).
- **Popular**[43, 29]: State + $k$ most popular actions from other agents (by followers, likes, etc.).
- **SFT**[30]: Supervised fine-tuning on state–action pairs; trained to convergence (Appendix Fig. 6).

For evaluation, we compare the outputs of our MF-LLM and baselines with ground-truth responses from the WEIBO corpus (hereafter **Real data**).

## 4.2 Evaluation Metrics

To evaluate the accuracy of LLM-based simulation of population decision-making dynamics, we adopt a micro-to-macro evaluation approach: (1) we first assess **individual** agent decisions (both real and generated actions); (2) then evaluate the **decision distribution** over a subset of agents at each timestep, capturing **temporal trends** in decision distribution over time. Accordingly, we design two types of metrics: one for individual actions and one for action distribution similarity.

**Metrics for Individual Actions.** To assess the policy model's ability to generate realistic actions for individual agents, we use GPT-4o-mini[3] to evaluate both real and generated text actions. We define evaluation dimensions such as *sentiment* (e.g., happy, angry, calm, doubtful), *attitude*, *behavior*, *stance*, *belief*, *subjectivity*, *intent*, and *rumor*, as informed by related work [29, 12]. A detailed list of evaluation dimensions, prompt templates, and the reliability and efficiency analysis of GPT-4o-mini as an evaluator are provided in Appendix F.3.

**Metrics for Action Distribution.** Building on individual action evaluation, we compute the action distribution over $N_t$ text actions at each timestep $t$. To assess the similarity between generated and real distributions, we use: *(1) KL Divergence* and *(2) Wasserstein Distance*: measure distributional similarity, averaged across all timesteps; *(3) Dynamic Time Warping (DTW) Distance*: evaluates temporal alignment between time series, assessing how well generated trends follow real dynamics; *(4) Negative Log-Likelihood (NLL) Loss*: measures log-probability of ground-truth actions; *(5) Macro F1* and *(6) Micro F1*: measure classification accuracy over generated action sets.

---

[2]https://www.kaggle.com/datasets/jackksoncsie/famous-keyword-twitter-replies-dataset/data

[3]We use the version o4-mini-2025-04-16.

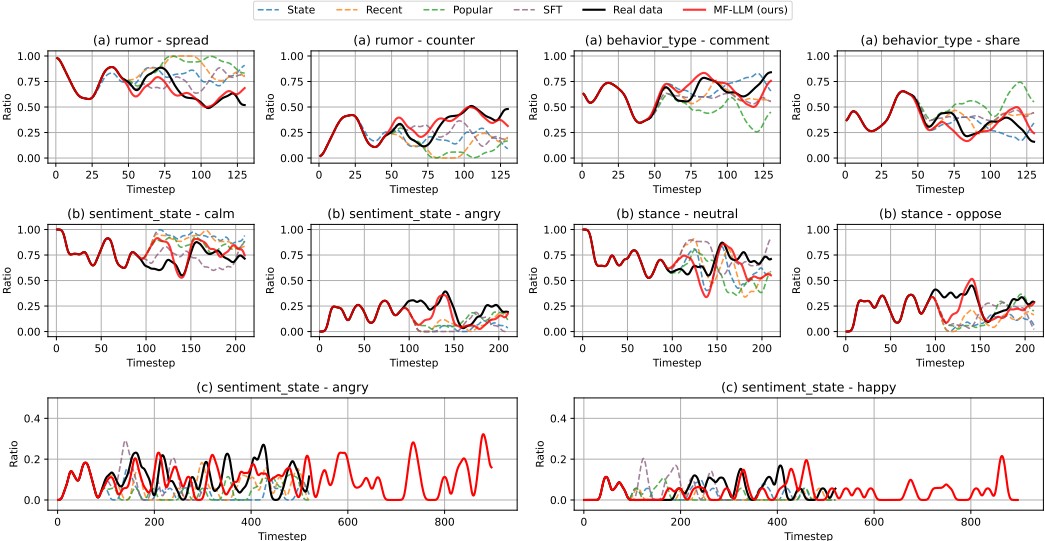

Figure 2: Comparison of collective decision trajectories in three events: our MF-LLM, *Real data*, and baselines—*State*, *Recent*, *Popular*, *SFT*. **Event (a)** Liu Xiang rumors (top row): spread vs. counter-rumor rates and comment vs. share behaviors. **Event (b)** Delayed retirement debate (middle row): calm vs. angry sentiment and neutral vs. opposing stances. **Event (c)** Weibo speech-freedom debate (bottom row): angry vs. happy sentiment over 900 timesteps (*Real data* spans 500). Our MF-LLM closely matches *Real data* trends in all events. Full curves see Appendix Fig. 7.

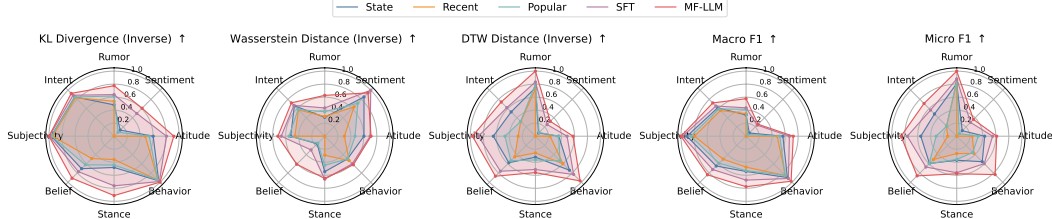

Figure 3: Radar plot comparing four baselines (*State*, *Recent*, *Popular*, *SFT*), and our MF-LLM (red)—on 5 distributional metrics and 8 semantic dimensions of actions. KL Divergence (Inverse), Wasserstein Distance (Inverse), and DTW Distance (Inverse) are inversely normalized so larger values indicate better performance, alongside Macro F1 and Micro F1. Larger areas denote superior performance, with MF-LLM (red) showing the **highest** semantic fidelity.

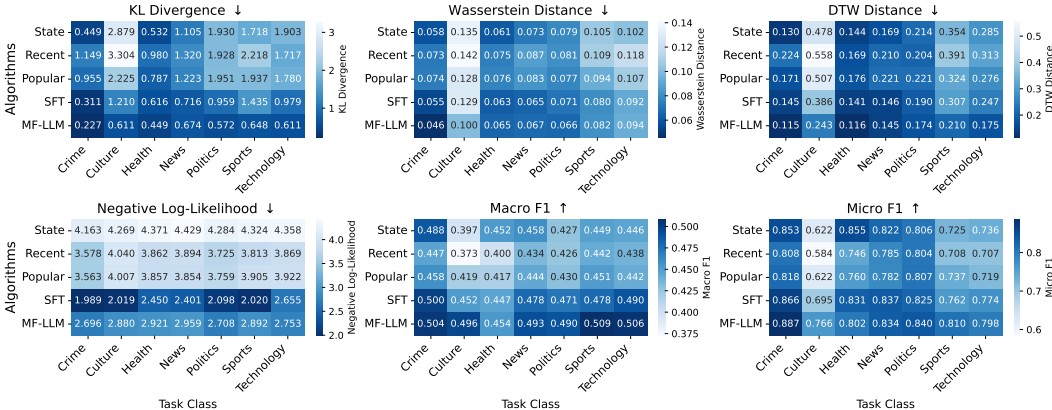

Figure 4: Comparison of algorithms across seven event domains. Heatmaps show six metrics evaluating action distribution fidelity, with **darker blue** indicating better performance. MF-LLM (ours) demonstrates strong generalization across all domains, even in challenging cases like *Culture*.

## 4.3 How Real is MF-LLM? Evaluating Its Fidelity and Generalization

**Facet A: Temporal Fidelity — Matching Real-World Decision Trajectories**  To demonstrate MF-LLM's ability to reproduce realistic decision trajectories over time, we simulate three events

with increasing horizons and agent scale: (a) short-horizon rumor propagation, (b) mid-horizon retirement debate, and (c) long-horizon speech freedom discourse (Fig. 2). For each event, we track key action metrics relevant to its topic (full curves in Appendix B Fig. 7). Across all events, MF-LLM (red) closely aligns with ground-truth data (black), while baselines—*State*, *Recent*, *Popular*, and *SFT*—often lag behind or smooth over critical transitions. **(a) Rumor Propagation.** MF-LLM accurately captures the tipping point around step 75, followed by a realistic decline. Baselines over-predict virality and fail to reverse the trend. **(b) Retirement Debate.** MF-LLM matches the anger peak at $T \approx 140$ within three steps, while *Popular* and *SFT* lag by over 20 steps, missing the timing of sentiment shifts. **(c) Speech Freedom.** Beyond matching the real-world trajectory up to $T \approx 500$, MF-LLM continues generating coherent trends up to $T = 900$ by resampling states from historical distributions (see Appendix G.2 for details). This ability arises from the iterative generation between the mean-field and policy models, enabling natural long-horizon simulation. A detailed analysis of the scalability and computational complexity of MF-LLM, from both theoretical and empirical perspectives, is provided in Appendix E.

**Facet B: Semantic Fidelity — Capturing the Meaning of Collective Behavior**  Beyond trajectory alignment, we assess MF-LLM's ability to model the semantic structure of collective decisions. We evaluate 20 real-world events involving up to 300 agents, measuring fidelity across five distributional metrics and eight semantic dimensions of actions (Fig. 3). We exclude NLL Loss as it lacks semantic interpretability. To ensure fair comparison, distance-based metrics (KL, Wasserstein, DTW) are inversely normalized so that higher values indicate better alignment. MF-LLM (red) consistently achieves the **largest** radar area, indicating stronger semantic fidelity across all dimensions. A clear performance hierarchy emerges: MF-LLM outperforms all baselines, followed by *SFT*, with *Popular*, *Recent*, and *State* trailing behind. While *SFT* performs better than other baselines on semantic metrics, it fails to track temporal shifts (see Fig. 2). In contrast, MF-LLM excels in both aspects, particularly on complex dimensions such as *sentiment* and *stance*, underscoring its strength in semantic decision modeling. See Appendix Table 4 for full results.

**Facet C: Cross-Domain Generalization — Transferring Across Diverse Event Domains**  Having validated fidelity, we next test MF-LLM's generalization across diverse event domains without task-specific adaptation. The test events span seven domains—*Crime*, *Culture*, *Health*, *News*, *Politics*, *Sports*, and *Technology*—each exhibiting distinct collective behaviors. As shown in Fig. 4, MF-LLM consistently lowers KL divergence (e.g., *Culture* $2.88 \rightarrow 0.61$) and DTW distance (e.g., *Sports* $0.35 \rightarrow 0.21$), while boosting classification performance in Micro and Macro F1. Although *SFT* achieves the lowest NLL loss, MF-LLM sacrifices marginal one-step accuracy for improved temporal and semantic consistency—precisely what KL, DTW, and F1 capture. Notably, MF-LLM remains robust even in complex domains like *Culture*, where other baselines struggle.

**Facet D: Backbone Robustness — Generalizing Across Base LLMs**  The previous experiments, based on `Qwen2-1.5B-Instruct`, validated MF-LLM's fidelity and generalization. We now extend MF-LLM to additional backbones—`GPT-4o-mini`, `DeepSeek-R1 Distill-Qwen-32B`, and `Qwen2-7B-Instruct` (Table 1). Since some LLMs are closed-source, we evaluate MF-LLM without fine-tuning. Even without tuning, MF-LLM consistently outperforms *State*, *Recent*, and *Popular* across all LLM backbones. Relative gains are most pronounced on `GPT-4o-mini`, with KL divergence reduced by **60**% and DTW by **21**%. `Qwen2-1.5B-Instruct` yields the best absolute performance. Here, MF-LLM achieves $33\%$ (KL) and $8\%$ (DTW) improvements, which further increase to **47**% and **16.8**% with IB-Tune. Interestingly, smaller models outperform larger ones in simulating collective decision dynamics, with detailed analysis in Section D. Overall, results highlight MF-LLM's robustness across diverse base LLMs—even without any model-specific adaptation.

## 4.4 What Drives MF-LLM? Dissecting the Mean Field and IB-Tune Mechanisms

**Ablation variants.** In §4.4, we evaluate six variants to assess the role of each MF-LLM component:

- **MF-LLM (ours)**: full model with both modules fine-tuned via IB-Tune.
- **w/o IB-Tune MF**: replace IB-Tune mean field model with pretrained `Qwen2-1.5B-Instruct`.
- **w/o IB-Tune Policy**: replace IB-Tune policy with pretrained; mean-field remains fine-tuned.
- **w/o IB-Tune**: both modules use pretrained LLM without fine-tuning.
- **SFT (no MF)**: drop mean-field module; policy trained via supervised fine-tuning.
- **Pretrained (no MF)**: drop mean-field module; use pretrained policy without fine-tuning.

Table 1: Comparison of base LLMs and baselines across multiple metrics. Lower is better (↓) for all metrics except Micro/Macro-F1 (↑). Values denote mean with improvement rate (%) relative to *State*. Cells with positive improvement rates are colored by magnitude.

| Backbone LLMs | KL Div. ↓ | Wass. Dist. ↓ | DTW ↓ | Macro F1 ↑ | Micro F1 ↑ | NLL Loss ↓ |
|---|---|---|---|---|---|---|
| `GPT-4o-mini` | | | | | | |
| **State** | 4.172 | 0.127 | 0.420 | 0.337 | 0.653 | – |
| **Recent** | 6.728 (-61.169%) | 0.145 (-14.173%) | 0.485 (-15.476%) | 0.270 (-19.881%) | 0.625 (-4.288%) | – |
| **Popular** | 5.591 (-34.037%) | 0.128 (-0.787%) | 0.447 (-6.429%) | 0.316 (-6.232%) | 0.650 (-0.459%) | – |
| **MF-LLM (ours)** | 1.647 (60.523%) | 0.100 (21.260%) | 0.329 (21.667%) | 0.399 (18.398%) | 0.724 (10.873%) | – |
| `DeepSeek-R1-32B` | | | | | | |
| **State** | 1.946 | 0.110 | 0.348 | 0.398 | 0.691 | – |
| **Recent** | 6.435 (-230.680%) | 0.141 (-28.182%) | 0.495 (-42.241%) | 0.281 (-29.397%) | 0.617 (-10.709%) | – |
| **Popular** | 3.812 (-95.890%) | 0.113 (-2.727%) | 0.385 (-10.632%) | 0.353 (-11.307%) | 0.682 (-1.302%) | – |
| **MF-LLM (ours)** | 1.280 (34.224%) | 0.088 (20.000%) | 0.307 (11.782%) | 0.418 (5.025%) | 0.724 (4.775%) | – |
| `Qwen2-7B-Instruct` | | | | | | |
| **State** | 1.153 | 0.085 | 0.238 | 0.436 | 0.773 | 4.175 |
| **Recent** | 1.346 (-16.739%) | 0.081 (4.706%) | 0.218 (8.403%) | 0.434 (-0.459%) | 0.780 (0.906%) | 4.183 (-0.192%) |
| **Popular** | 1.066 (7.545%) | 0.076 (10.588%) | 0.199 (16.387%) | 0.445 (2.064%) | 0.794 (2.717%) | 4.177 (-0.048%) |
| **MF-LLM (ours)** | 1.010 (12.402%) | 0.075 (11.765%) | 0.198 (16.807%) | 0.447 (2.523%) | 0.796 (2.975%) | 4.132 (1.030%) |
| `Qwen2-1.5B-Instruct` | | | | | | |
| **State** | 0.966 | 0.068 | 0.166 | 0.463 | 0.823 | 4.138 |
| **Recent** | 1.277 (-32.192%) | 0.077 (-13.235%) | 0.202 (-21.687%) | 0.441 (-4.752%) | 0.799 (-2.917%) | 4.056 (1.981%) |
| **Popular** | 1.288 (-33.333%) | 0.080 (-17.647%) | 0.199 (-19.880%) | 0.435 (-6.048%) | 0.792 (-3.767%) | 4.080 (1.402%) |
| **MF-LLM (ours)** | 0.645 (33.230%) | 0.065 (4.412%) | 0.152 (8.434%) | 0.486 (4.968%) | 0.839 (1.944%) | 4.085 (1.281%) |
| **MF-LLM IB-Tune (ours)** | **0.512** (47.002%) | **0.062** (8.824%) | **0.138** (16.867%) | **0.495** (6.911%) | **0.846** (2.795%) | **2.809** (32.091%) |

Table 2: Ablation study on MF-LLM. We evaluate the contribution of the mean-field module and IB-Tune by comparing six variants. Cells show metric values and relative changes from MF-LLM (ours); darker shading indicates larger drops, reflecting the importance of the removed component.

| Method | KL Div. ↓ | Wass. Dist. ↓ | DTW Dis. ↓ | NLL Loss ↓ | Macro F1 ↑ | Micro F1 ↑ |
|---|---|---|---|---|---|---|
| **MF-LLM (Ours)** | **0.4813** | **0.0703** | **0.1581** | 2.9029 | **0.4891** | **0.8303** |
| **w/o IB-Tune MF** | 0.5312 (10.4%) | 0.0702 (-0.1%) | 0.1586 (0.3%) | 2.9802 (2.7%) | 0.4865 (-0.5%) | 0.8284 (-0.2%) |
| **w/o IB-Tune Policy** | 0.5882 (22.2%) | 0.0704 (0.1%) | 0.1582 (0.1%) | 4.0454 (39.4%) | 0.4846 (-0.9%) | 0.8300 (-0.0%) |
| **w/o IB-Tune** | 0.6166 (28.1%) | 0.0716 (1.8%) | 0.1636 (3.5%) | 4.0933 (41.0%) | 0.4828 (-1.3%) | 0.8292 (-0.1%) |
| **SFT (no MF)** | 0.7531 (56.5%) | 0.0746 (6.1%) | 0.1859 (17.5%) | 2.4217 (-16.6%) | 0.4755 (-2.8%) | 0.8164 (-1.7%) |
| **Pretrained (no MF)** | 1.0526 (118.7%) | 0.0792 (12.7%) | 0.1946 (23.1%) | 4.2088 (45.0%) | 0.4561 (-6.7%) | 0.8076 (-2.7%) |

**Mean-Field Module Is Key to Population Dynamics.** Table 2 demonstrates the critical role of the mean-field module in achieving high-fidelity simulations. Removing it increases KL divergence by 118% and reduces Macro F1 by up to 7%. Although *SFT (no MF)* yields the lowest NLL Loss (2.4217), its KL divergence exceeds ours by over 50%, suggesting that low token-level loss alone may be insufficient for capturing social dynamics. These results confirm that the dynamic population signal captured by MF-LLM is essential for reproducing realistic decision trajectories over time.

**IB-Tune Enhances Real-World Alignment.** From Table 2, **(i) Removing IB-Tune** from both the mean-field and policy models (*w/o IB-Tune*) increases KL by 28.1% and NLL Loss by 41.0% relative to *MF-LLM*, demonstrating that combined fine-tuning is essential. **(ii) Isolating IB-tuned mean-field** (*MF-LLM* vs. *w/o IB-Tune MF*) cuts KL by 10.4% and NLL Loss by 2.7%, showing that IB objective is key to aligning population trends and suppressing noise. **(iii) Tuning Policy model via IB-Tune** yields large gains: it lowers KL by up to 22.2% and NLL Loss by up to 39.4% across variants, further boosting distributional fidelity. Together, these results highlight IB-Tune's two-stage role: refining the mean field for population alignment, then calibrating the policy for better action likelihoods—both essential for aligning MF-LLM with real-world decisions.

## 5 Discussion and Conclusion

**Key Findings** from our experiments: **(1)** MF-LLM accurately captures real-world population decision dynamics, both in decision trajectories and semantic fidelity. **(2)** It generalizes well across diverse event domains and LLM backbones. **(3)** The *mean-field module* is the key for capturing population dynamics, while *IB-Tune* greatly enhances alignment with real-world data.

**Applications and Discussion.** Inspired by these findings, we explore MF-LLM's broader potential (Appendix D): **(1)** Enables forecasting of public opinion and pre-evaluation of intervention effects

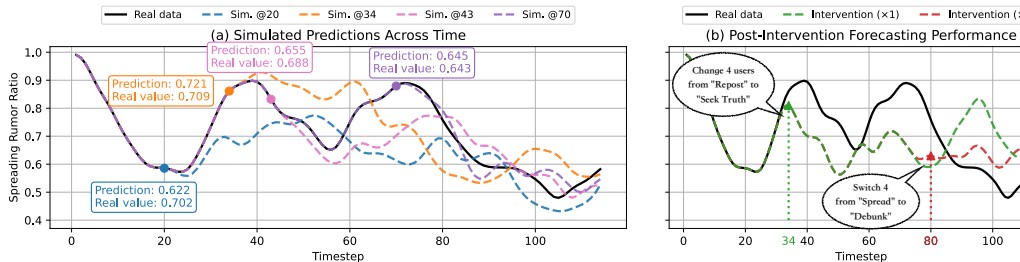

Figure 5: MF-LLM for prediction and intervention: (a) Rumor spread predicted from four start points (20, 34, 43, 70); later starts improve accuracy. (b) Predictions under two interventions: single intervention at step 34 (with later rebound); second intervention at step 80 yields desired effect.

(Fig. 5). **(2)** Injecting exogenous signals improves alignment with real-world dynamics (Fig. 8). We further observe: **(3)** Low NLL does not necessarily imply high rollout fidelity. **(4)** Smaller LLMs better capture agent diversity than larger ones in collective simulations.

**Conclusion.** MF-LLM integrates mean-field theory with LLMs to model the interplay between individual decisions and population dynamics, enabling high-fidelity simulation of collective behavior over time. IB-Tune further improves alignment with real-world data. These results suggest that uncovering collective behavior principles, combined with data-driven fine-tuning, provides a strong foundation for accurate and scalable social simulation.

# Acknowledgments

We would like to thank Prof. Bo Li from Peking University for his valuable discussions and suggestions. This work was supported by the National Natural Science Foundation of China (NSFC) under the Original Exploration Program, Grant No. 72450002. We also thank the Program Chairs, Senior Area Chairs, Area Chairs, and Reviewers for their constructive feedback.

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

# Appendix

**Appendix Table of Contents**

## A   Training Curves and Hyperparameters

We present the training loss curves for both the policy model and the mean field model optimized via our IB-Tune algorithm, along with the loss curve of the LLM trained using standard supervised fine-tuning (SFT), in Figure 6. All models are fine-tuned based on the `Qwen2-1.5B-Instruct` language model. The corresponding training hyperparameters are summarized in Table 3. All experiments were conducted on a machine equipped with two NVIDIA A100-PCIE-40GB GPUs (CUDA 12.2, driver version 535.183.01), each with 40GB of memory. The mean field model was trained for approximately 74 hours, the policy model for 64 hours, and the SFT model for 25 hours.

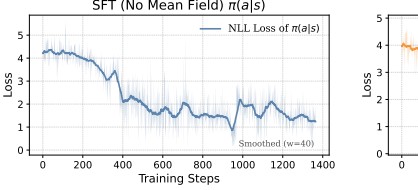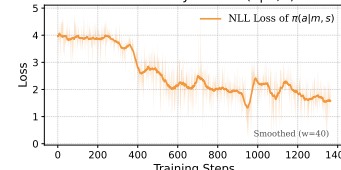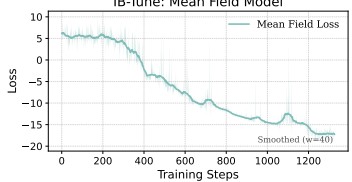

Figure 6: Training loss curves for three LLMs (Qwen2-1.5B): (left) SFT (No Mean Field), representing standard supervised fine-tuning on state–action pairs; (middle) IB-Tune: Policy Model $\pi(a|m,s)$, corresponding to the fine-tuning of the policy model via our IB-Tune method; and (right) IB-Tune: Mean Field Model, corresponding to the fine-tuning of the mean field model via IB-Tune. The loss functions for the latter two models are detailed in Section 3.3. Each curve shows the moving average smoothed loss (window size 40), with shaded areas indicating the original unsmoothed losses. All models are fine-tuned to convergence.

Table 3: Training hyperparameters for IB-Tune (the mean field model, the policy model), and the standard SFT algorithm. All models are fine-tuned from `Qwen2-1.5B-Instruct`.

| Hyperparameter | Mean Field Model | Policy Model | Standard SFT |
|---|---|---|---|
| Base model | `Qwen2-1.5B-Instruct` | `Qwen2-1.5B-Instruct` | `Qwen2-1.5B-Instruct` |
| Max sequence length | 2048 | 2048 | 2048 |
| Training dataset | Weibo | Weibo | Weibo |
| Training events (until converge) | 400 | 400 | 400 |
| Training batch size | 256 | 256 | 256 |
| Micro batch size | 8 | 8 | 8 |
| Max epochs | 1 | 1 | 1 |
| Learning rate | $5 \times 10^{-7}$ | $5 \times 10^{-7}$ | $1 \times 10^{-6}$ |
| LoRA rank | 64 | 64 | 64 |
| LoRA alpha | 64 | 64 | 64 |
| Zero stage | 2 | 2 | 2 |
| Loss function | $\mathcal{L}_{\text{mean field}}$ of IB-Tune | $\mathcal{L}_{\text{policy}}$ of IB-Tune | NLL Loss of $\pi(a|s)$ |
| $\beta$ in $\mathcal{L}_{\text{mean field}}$ | 2 | - | - |

# B  Additional Experimental Results

## B.1  Additional Results on WEIBO corpus

**Results across Different Decision Dimensions.**    In Table 4, we first present the similarity metrics comparing our MF-LLM method against the baselines State, Recent, Popular, and SFT across each decision dimension (base LLM is Qwen2-1.5B-Instruct). At each evaluation step, the action distribution is constructed from the actions of the 16 nearest agents. Similarity is measured between the real and generated distributions using multiple metrics, including KL divergence, Wasserstein distance, DTW, Macro F1, Micro F1, and NLL Loss. The reported results are averaged over 14 task scenarios. For each task, evaluation is performed by simulating agents from index 50 to 300.

Table 4: Evaluation results across different decision dimensions. All metrics are reported with relative improvements (%) over State. **MF-LLM (ours)** consistently achieves the best performance.

| Algorithm | KL Div. ↓ | Wass. Dist. ↓ | DTW ↓ | Macro F1 ↑ | Micro F1 ↑ | NLL Loss ↓ |
|---|---|---|---|---|---|---|
| **Dimension: Rumor** | | | | | | |
| **State** | 2.2057 | 0.1061 | 0.1741 | 0.3731 | 0.8387 | 4.3154 |
| **Recent** | 1.9807 (10.2%) | 0.1050 (1.0%) | 0.1813 (-4.1%) | 0.3794 (1.7%) | 0.8399 (0.1%) | 3.8282 (11.3%) |
| **Popular** | 1.6483 (25.3%) | 0.0993 (6.4%) | 0.1584 (9.0%) | 0.3997 (7.1%) | 0.8485 (1.2%) | 3.8379 (11.1%) |
| **SFT (No MF)** | 1.5597 (29.3%) | 0.0948 (10.6%) | 0.1459 (16.2%) | 0.4060 (8.8%) | 0.8536 (1.8%) | 2.2280 (48.4%) |
| **MF-LLM (ours)** | **1.0032** (54.5%) | **0.0796** (25.0%) | **0.0926** (46.8%) | **0.4481** (20.1%) | **0.8784** (4.7%) | **2.8342** (34.3%) |
| **Dimension: Sentiment** | | | | | | |
| **State** | 3.6245 | 0.0619 | 0.3895 | 0.2960 | 0.6960 | 4.3154 |
| **Recent** | 4.1686 (-15.0%) | 0.0789 (-27.5%) | 0.4120 (-5.8%) | 0.2750 (-7.1%) | 0.6711 (-3.6%) | 3.8282 (11.3%) |
| **Popular** | 3.8127 (-5.2%) | 0.0724 (-16.9%) | 0.3862 (0.8%) | 0.2862 (-3.3%) | 0.6824 (-2.0%) | 3.8379 (11.1%) |
| **SFT (No MF)** | 2.1504 (40.7%) | 0.0504 (18.6%) | 0.3310 (15.0%) | 0.3473 (17.3%) | 0.7332 (5.4%) | 2.2280 (48.4%) |
| **MF-LLM (ours)** | **1.6850** (53.5%) | **0.0550** (11.2%) | **0.3087** (20.7%) | **0.3530** (19.3%) | **0.7464** (7.2%) | **2.8342** (34.3%) |
| **Dimension: Attitude** | | | | | | |
| **State** | 1.7155 | 0.0816 | 0.2973 | 0.4449 | 0.7601 | 4.3154 |
| **Recent** | 2.2148 (-29.1%) | 0.1052 (-28.9%) | 0.3399 (-14.4%) | 0.4185 (-5.9%) | 0.7204 (-5.2%) | 3.8282 (11.3%) |
| **Popular** | 1.9632 (-14.5%) | 0.0923 (-13.1%) | 0.2983 (-0.3%) | 0.4303 (-3.3%) | 0.7396 (-2.7%) | 3.8379 (11.1%) |
| **SFT (No MF)** | 0.8818 (48.6%) | 0.0741 (9.2%) | 0.2566 (13.7%) | 0.4750 (6.8%) | 0.7854 (3.3%) | 2.2280 (48.4%) |
| **MF-LLM (ours)** | **0.4245** (75.3%) | **0.0730** (10.5%) | **0.2272** (23.6%) | **0.4928** (10.8%) | **0.7983** (5.0%) | **2.8342** (34.3%) |
| **Dimension: Behavior** | | | | | | |
| **State** | 0.2132 | 0.0901 | 0.1758 | 0.5446 | 0.7859 | 4.3154 |
| **Recent** | 0.2756 (-29.3%) | 0.0873 (3.1%) | 0.2208 (-25.6%) | 0.5317 (-2.4%) | 0.7489 (-4.7%) | 3.8282 (11.3%) |
| **Popular** | 0.2963 (-38.9%) | 0.0901 (0.0%) | 0.2411 (-37.1%) | 0.5298 (-2.7%) | 0.7437 (-5.4%) | 3.8379 (11.1%) |
| **SFT (No MF)** | 0.1505 (29.4%) | 0.0819 (9.1%) | 0.1467 (16.5%) | 0.5511 (1.2%) | 0.7948 (1.1%) | 2.2280 (48.4%) |
| **MF-LLM (ours)** | **0.0956** (55.2%) | **0.0802** (11.0%) | **0.1016** (42.2%) | **0.5688** (4.4%) | **0.8440** (7.4%) | **2.8342** (34.3%) |
| **Dimension: Stance** | | | | | | |
| **State** | 2.2091 | 0.0861 | 0.3092 | 0.4376 | 0.7477 | 4.3154 |
| **Recent** | 2.7059 (-22.5%) | 0.1059 (-23.0%) | 0.3302 (-6.8%) | 0.4177 (-4.5%) | 0.7263 (-2.9%) | 3.8282 (11.3%) |
| **Popular** | 2.3616 (-6.9%) | 0.0938 (-9.0%) | 0.2955 (4.4%) | 0.4339 (-0.8%) | 0.7450 (-0.4%) | 3.8379 (11.1%) |
| **SFT (No MF)** | 1.0596 (52.0%) | 0.0771 (10.5%) | 0.2485 (19.6%) | 0.4776 (9.1%) | 0.7874 (5.3%) | 2.2280 (48.4%) |
| **MF-LLM (ours)** | **0.4489** (79.7%) | **0.0779** (9.5%) | **0.2335** (24.5%) | **0.5080** (16.1%) | **0.7907** (5.7%) | **2.8342** (34.3%) |
| **Dimension: Belief** | | | | | | |
| **State** | 1.2888 | 0.1159 | 0.2279 | 0.4695 | 0.7939 | 4.3154 |
| **Recent** | 2.1881 (-69.8%) | 0.1288 (-11.1%) | 0.2685 (-17.8%) | 0.4247 (-9.5%) | 0.7763 (-2.2%) | 3.8282 (11.3%) |
| **Popular** | 1.6316 (-26.6%) | 0.1170 (-1.0%) | 0.2416 (-6.0%) | 0.4516 (-3.8%) | 0.7948 (0.1%) | 3.8379 (11.1%) |
| **SFT (No MF)** | 1.0178 (21.0%) | 0.1055 (9.0%) | 0.1691 (25.8%) | 0.4927 (4.9%) | 0.8100 (2.0%) | 2.2280 (48.4%) |
| **MF-LLM (ours)** | **0.4432** (65.6%) | **0.0811** (30.0%) | **0.1345** (41.0%) | **0.5270** (12.2%) | **0.8498** (7.0%) | **2.8342** (34.3%) |
| **Dimension: Subjectivity** | | | | | | |
| **State** | 0.2348 | 0.0878 | 0.2054 | 0.5481 | 0.7857 | 4.3154 |
| **Recent** | 0.3237 (-37.8%) | 0.0929 (-5.8%) | 0.3271 (-59.3%) | 0.5130 (-6.4%) | 0.6999 (-10.9%) | 3.8282 (11.3%) |
| **Popular** | 0.2542 (-8.3%) | 0.0858 (2.3%) | 0.2627 (-27.9%) | 0.5299 (-3.3%) | 0.7369 (-6.2%) | 3.8379 (11.1%) |
| **SFT (No MF)** | 0.1097 (53.3%) | 0.0764 (13.0%) | 0.1326 (35.4%) | 0.5678 (3.6%) | 0.8286 (5.5%) | 2.2280 (48.4%) |
| **MF-LLM (ours)** | **0.0891** (62.1%) | **0.0724** (17.5%) | **0.1030** (49.8%) | **0.5754** (5.0%) | **0.8484** (8.0%) | **2.8342** (34.3%) |
| **Dimension: Intent** | | | | | | |
| **State** | 0.6812 | 0.0770 | 0.2435 | 0.4602 | 0.7712 | 4.3154 |
| **Recent** | 0.7546 (-10.8%) | 0.0841 (-9.2%) | 0.3100 (-27.3%) | 0.4424 (-3.9%) | 0.7174 (-7.0%) | 3.8282 (11.3%) |
| **Popular** | 0.6901 (-1.3%) | 0.0834 (-8.3%) | 0.3096 (-27.2%) | 0.4525 (-1.7%) | 0.7164 (-7.1%) | 3.8379 (11.1%) |
| **SFT (No MF)** | 0.5381 (21.0%) | 0.0763 (0.9%) | 0.2137 (12.3%) | 0.4719 (2.5%) | 0.7889 (2.3%) | 2.2280 (48.4%) |
| **MF-LLM (ours)** | **0.3714** (45.5%) | **0.0720** (6.5%) | **0.1747** (28.2%) | **0.4921** (6.9%) | **0.8185** (6.1%) | **2.8342** (34.3%) |

**Full Dynamic Trend Simulations.** In Appendix Fig. 7, we present the full dynamic trend simulations under the *Liu Xiang Rumor* scenario. We illustrate the dynamic trends across all decision dimensions and compare the simulations of multiple baselines, including `State`, `Recent`, `Popular`, and `SFT`. Additionally, we include simulation curves for the ablation variants of our proposed `MF-LLM`. Due to the large number of baselines and evaluation dimensions, we report these detailed results in the appendix for clarity.

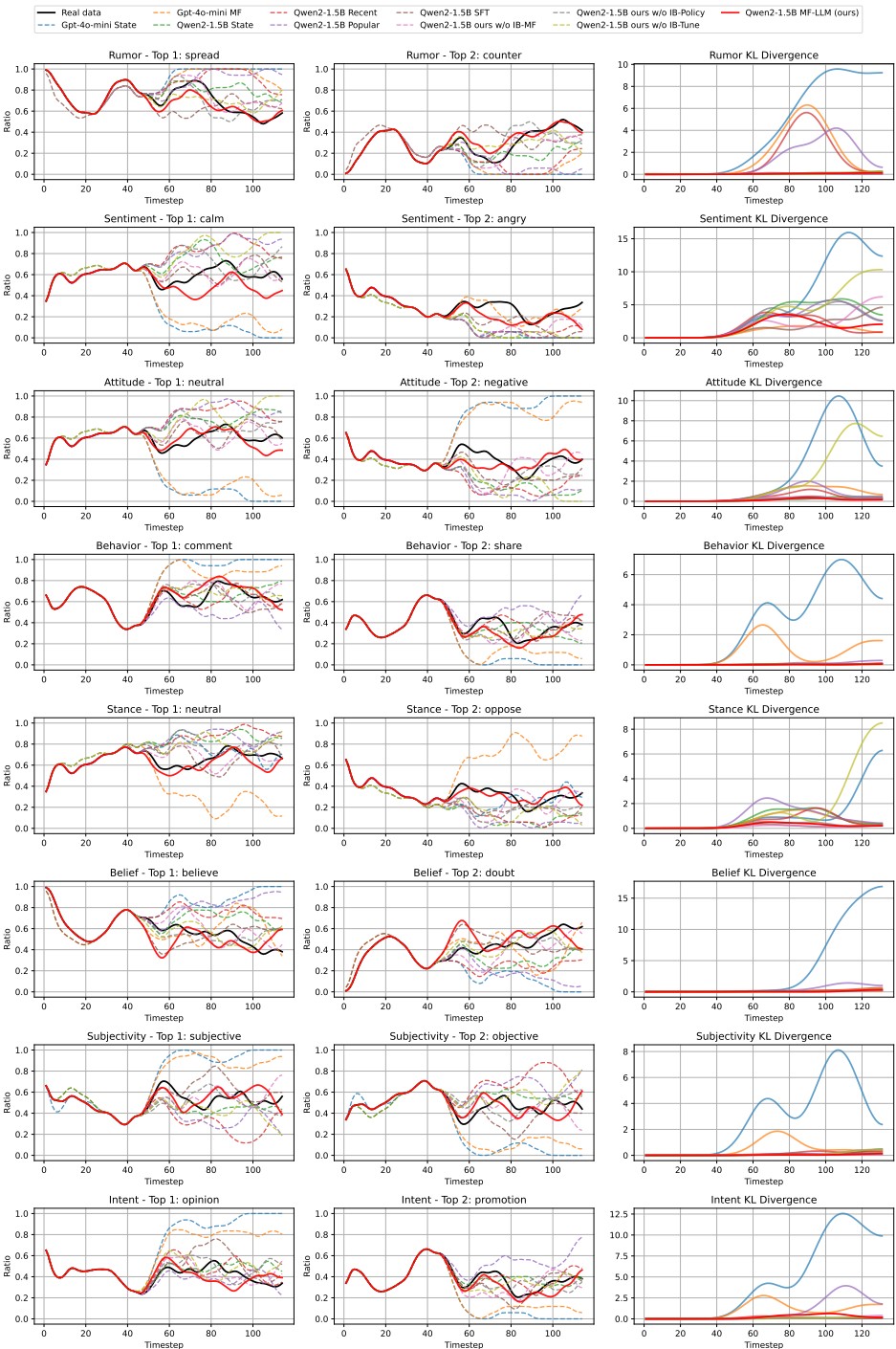

Figure 7: Comparison of dynamic trend simulations across multiple baselines (starting from step 48). At each timestep, the distribution of actions is estimated based on the actions of the 16 nearest agents.

## B.2 Additional Results on TWITTER dataset

We extend evaluation beyond the WEIBO corpus to examine whether MF-LLM preserves distributional fidelity and decision consistency on other datasets featuring different interaction patterns.

**Famous-Keyword Twitter Replies Dataset (Kaggle).** The TWITTER dataset[4] contains a large collection of tweet–reply pairs centered on popular keywords, providing a suitable testbed for large-scale discussion dynamics. Testing on several base LLMs, MF-LLM outperforms baselines without any domain-specific fine-tuning:

Table 5: Results on the FAMOUS-KEYWORD TWITTER REPLIES dataset across different LLM backbones. Lower is better (↓) for all metrics except Micro/Macro-F1 (↑). MF-LLM consistently generalizes across backbones without domain-specific fine-tuning.

| Backbone LLMs | KL Div. ↓ | Wass. Dist. ↓ | DTW ↓ | Macro F1 ↑ | Micro F1 ↑ | NLL Loss ↓ |
|---|---|---|---|---|---|---|
| `Qwen2-1.5B-Instruct` | | | | | | |
| **State** | 0.8696 | **0.0673** | 0.1700 | 0.4977 | **0.8241** | 4.2121 |
| **Recent** | 1.5061 | 0.0922 | 0.2206 | 0.4471 | 0.7661 | 4.1053 |
| **Popular** | 1.0059 | 0.0725 | **0.1553** | 0.4864 | 0.8047 | 4.0697 |
| **MF-LLM (ours)** | **0.6883** | 0.0744 | 0.1587 | **0.4989** | 0.8093 | **3.9748** |
| `GPT-4o-mini` | | | | | | |
| **State** | 6.8382 | **0.1469** | 0.4513 | 0.2808 | 0.6282 | – |
| **Recent** | 6.5273 | 0.1491 | 0.4444 | 0.2843 | 0.6361 | – |
| **Popular** | 7.5150 | 0.1634 | 0.4251 | 0.2526 | 0.6112 | – |
| **MF-LLM (ours)** | **5.9322** | 0.1506 | **0.3300** | **0.2934** | **0.6403** | – |
| `GPT-4o` | | | | | | |
| **State** | 7.8105 | 0.1711 | 0.5178 | 0.2388 | 0.5962 | – |
| **Recent** | 7.2690 | 0.1607 | 0.4576 | 0.2574 | 0.6200 | – |
| **Popular** | 8.8492 | 0.1818 | 0.5198 | 0.2138 | 0.5798 | – |
| **MF-LLM (ours)** | **2.2669** | **0.1085** | **0.2211** | **0.4103** | **0.7249** | – |
| `DeepSeek-R1-Distill-Qwen-32B` | | | | | | |
| **State** | 1.7496 | 0.1105 | 0.2847 | 0.4173 | 0.6999 | – |
| **Recent** | 2.4299 | 0.1192 | 0.3462 | 0.3941 | 0.6829 | – |
| **Popular** | 1.4311 | **0.0905** | 0.2198 | 0.4424 | **0.7256** | – |
| **MF-LLM (ours)** | **1.1814** | 0.0922 | **0.2042** | **0.4483** | 0.7204 | – |

**Market Behavior Simulation.** Following prior studies on market-style decision dynamics (e.g., *TwinMarket* [42], *ElectionSim* [44]), we construct a Bitcoin-related tweet–reply dataset to simulate *bullish/bearish* sentiment evolution. MF-LLM again achieves the best quantitative alignment:

Table 6: Results on the BITCOIN-RELATED MARKET BEHAVIOR SIMULATION dataset. Lower is better (↓) for all metrics except Micro/Macro-F1 (↑). MF-LLM achieves the best quantitative alignment across all metrics with base LLM `Qwen2-1.5B-Instruct`).

| Method | KL Div. ↓ | Wass. Dist. ↓ | DTW ↓ | Macro F1 ↑ | Micro F1 ↑ | NLL Loss ↓ |
|---|---|---|---|---|---|---|
| State | 1.2304 | 0.0854 | 0.4505 | 0.4723 | 0.6721 | 4.4320 |
| Recent | 2.0641 | **0.0800** | 0.4730 | 0.4550 | 0.6573 | 4.3438 |
| Popular | 1.4394 | 0.0853 | 0.3930 | 0.4834 | 0.6920 | 4.2537 |
| **MF-LLM (ours)** | **0.9316** | 0.0915 | **0.2364** | **0.5025** | **0.7741** | **4.1485** |

Across both datasets, MF-LLM maintains strong alignment in distributional metrics (KL, WD, DTW) and classification performance (Macro/Micro F1), indicating stable generalization under different social and linguistic contexts.

## C Modeling Complex and Heterogeneous Interactions in MF-LLM

MF-LLM naturally models complex and heterogeneous interactions among agents in large populations. This section provides an intuitive explanation of its mechanism and several illustrative examples from real-world data, showing how MF-LLM captures diverse influence patterns and topic-specific behaviors.

---

[4]https://www.kaggle.com/datasets/jackksoncsie/famous-keyword-twitter-replies-dataset/data

**Intuitive Mechanism.** In MF-LLM, the *policy model* maps the current population signal to individual actions, while the *mean field model* updates the population signal from newly observed actions. The pathway

$$a_1 \rightarrow m_{t+1} \rightarrow a_2$$

represents an indirect but generalizable interaction between any pair of agents. Unlike prior work with fixed influence weights, MF-LLM infers and updates interaction effects dynamically from online data, enabling robust modeling even in unseen structures.

**Empirical Illustration.** To better understand this mechanism, we present several examples demonstrating that MF-LLM effectively captures (1) heterogeneous interactions, (2) disproportionate influence, and (3) small close-knit groups, without manually defined interaction weights or structures.

**(1) Heterogeneous Interactions.** We compare two different actions taken by the same *high-influence user*.

---

**Example of Heterogeneous Interactions**

**State:** A non-verified user with high influence and many friends.
**Previous Mean Field:** Most users express opposition and criticism toward the event, regarding it as unfair competition.
- **Action A:** "Repost"
- **Action B:** "Looking forward to the truth!"

**Updated Mean Field:**
- After **Action A**: "Most users are criticizing the event rather than seeking factual verification."
- After **Action B**: "Users call for an investigation to uncover the cause. Many suspect that this might be a genuine news report."

---

**(2) Disproportionate Influence.** We compare the same action taken by users of different influence levels.

---

**Example of Disproportionate Influence**

**Action:** "Looking forward to the truth!"
**Previous Mean Field:** Most users express opposition and criticism toward the event, regarding it as unfair competition.
**Agent States:**
- **High-Influence User**: A non-verified user with high influence and many friends.
- **Low-Influence User**: A similar user with low influence and few friends.

**Updated Mean Field:**
- After **High-Influence User** acts: "Many suspect that this might be a genuine news report."
- After **Low-Influence User** acts: "A few users in the comments express a desire to uncover the truth."

---

**(3) Small, Close-Knit Groups.** Finally, we illustrate localized topic-specific interactions.

---

**Example of Small and Close-Knit Groups**

**Conversation Context:** In an entertainment discussion:
- **User A**: "Taylor Swift will hold a concert!"
- **User B**: "The city marathon route will be adjusted next month."

**Updated Mean Field (after A + B):** "Population sentiment surges around the concert... marathon news draws moderate interest from runners."
**Policy Model Responses under the Same Mean Field:**
- **Idol Fan User**: "Can't believe it! Already organizing a group to buy tickets and cheer!"
- **Marathon Enthusiast**: "Good to know about the route change—I'll adjust my training."

---

This shows MF-LLM captures topic-specific shifts via the mean field and generates heterogeneous actions via the policy model, faithfully modeling "close-knit group" dynamics from real data.

**Summary.**    These examples demonstrate that MF-LLM adaptively models heterogeneous and asymmetric agent behaviors through iterative mean-field updates.

# D    Applications and Discussion of MF-LLM

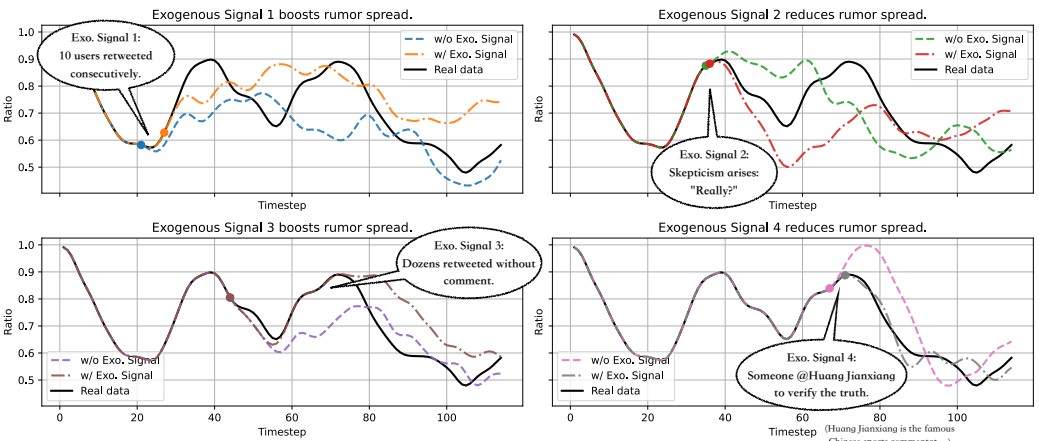

Figure 8: Simulation before or after key nodes. Key node 1 and 3 boost rumor spread, Key node 2 and 4 reduce rumor spread.

We first highlight three key findings from our experiments. **(1)** MF-LLM accurately **matches real-world decision dynamics**, in both temporal shifts and semantic structure. **(2)** It **generalizes** well across social domains and backbone LLMs without task-specific tuning. **(3)** The **mean-field module and IB-Tune are both essential**—removing either significantly degrades fidelity. These results validate MF-LLM as a robust framework for simulating collective behavior. Inspired by them, we now explore more practical applications and further discussions.

**Forecasting Public Opinion and Pre-evaluating Interventions.**    We demonstrate MF-LLM's utility in real-time forecasting and intervention design using a rumor propagation scenario (Fig. 5). The framework supports (i) trend forecasting, (ii) intervention planning, and (iii) evaluation of intervention outcomes. **(a) Forecasting accuracy.** Starting from partial observations at steps 20, 34, 43, and 70, MF-LLM generates forward rollouts of population behavior. Forecasts closely track ground-truth rumor prevalence (e.g., step 70: predicted 0.645 vs. actual 0.643), highlighting MF-LLM's temporal consistency and reliability. **(b) Intervention planning.** Simulation outputs enable data-driven planning. At step 34, the model anticipates a rise in rumor spread, triggering an early warning for intervention. This offers actionable insights for proactive decision-making. **(c) Intervention evaluation.** We compare two intervention strategies. A single intervention at step 34 (green) slows spread temporarily but fails to prevent a rebound. A second, later intervention at step 80 (red) stabilizes the trajectory. MF-LLM thus supports pre-deployment testing of multi-step interventions.

**Incorporating Exogenous Signals to Improve Fidelity.**    While MF-LLM captures endogenous dynamics driven by agent interactions, many real-world events are shaped by rare, high-impact exogenous signals. We investigate their effect using the Liu Xiang rumor scenario, where a burst of "silent reposts" triggers widespread misperception. By injecting such signals into the simulation, we reproduce observed inflection points (Fig. 8) and improve alignment with real-world dynamics. These findings underscore the importance of modeling rare, externally induced behaviors. Future work may explore dynamic event detection, uncertainty-aware responses, and automated signal injection to capture both endogenous patterns and exogenous shocks.

**Low NLL does not Necessarily Imply High Rollout Fidelity.** In Fig. 4 and Table 2, we observe that *SFT* achieves the lowest NLL Loss at test time. However, token-level likelihood does not necessarily translate to rollout-level fidelity. MF-LLM solves a richer conditional generation problem by aligning each decision with the evolving population signal $m$. This additional constraint increases irreducible cross-entropy slightly but prevents overfitting and encourages long-term semantic consistency. As a result, MF-LLM outperforms *SFT* on KL divergence, DTW distance, and F1 metrics. The trade-off is favorable: sacrificing minimal one-step accuracy yields substantial gains in rollout quality and realism.

**Smaller LLMs Outperform Larger Ones in Simulating Population Dynamics.** In §4.3, we find that smaller backbones such as `Qwen2-1.5B-Instruct` outperform larger models in simulating collective decision dynamics. Analysis of outputs from `GPT-4o-mini` and `DeepSeek-R1` reveals highly **homogeneous agent responses**, even under identical prompts and hyperparameters across different LLMs (Appendix I). Larger models tend to produce uniform responses over time, limiting their ability to capture diverse agent behavior and degrading long-horizon simulation fidelity. This likely stems from reduced sensitivity to subtle variations in prompt or agent state. In contrast, smaller models exhibit greater responsiveness and variability, resulting in more realistic and dynamic population trajectories. These findings align with prior work showing that large LLMs often favor high-probability outputs [18], and may underperform in tasks requiring structured behavioral diversity [47]. These results provide key insights for social simulation: reducing output homogeneity may be more important than maximizing model size, and properly leveraging smaller models can preserve behavioral diversity while offering substantial efficiency gains.

**Limitations and Future Work** While incorporating exogenous signals improves simulation fidelity, the current MF-LLM framework does not yet support real-time detection or integration of such signals. Future work could explore automated mechanisms for monitoring and injecting external events into the simulation loop. In addition, as simulation horizons increase, small deviations in early steps may compound over time, leading to distributional drift. This effect is observed in Fig. 2, and highlights the need for long-horizon correction strategies, such as periodic recalibration or error-aware feedback mechanisms.

## E   Scalability and Complexity Analysis of MF-LLM

**Computational Complexity.** MF-LLM aims to capture dynamic agent interactions that give rise to collective behavior. Modeling all pairwise agent interactions would incur $\mathcal{O}(N^2)$ computational cost, whereas MF-LLM reduces this to $\mathcal{O}(N)$ through a mean-field approximation. Specifically, the policy model generates one personalized action per agent, resulting in $\mathcal{O}(N)$ LLM inferences, and the mean-field model updates the population signal once per batch of $K$ agents, producing $\mathcal{O}(N/K)$ mean-field inferences. Overall, MF-LLM achieves **linear-time complexity** while still modeling inter-agent effects and reproducing population-level dynamics. This contrasts with baseline simulators that either ignore agent interactions entirely or approximate them through manual prompt engineering without dynamic coupling.

**Empirical Validation of Complexity.** To empirically validate the theoretical $\mathcal{O}(N)$ analysis, we conduct two complementary experiments: (i) scalability testing under larger agent populations and (ii) runtime under increasing simulation scales.

**(a) Scalability under Larger Populations.** We evaluate MF-LLM's scalability by extending simulations up to **4,000 agents** using `Qwen2-1.5B-Instruct` (without fine-tuning). Results in Table 7 show that MF-LLM maintains consistent accuracy as population size increases, demonstrating stable performance and convergence in large-scale settings.

**(b) Runtime Scaling.** We further measure runtime on `Qwen2-1.5B-Instruct` using the TWITTER dataset (no fine-tuning) on a single NVIDIA A100 GPU (40 GB memory). As shown in Table 8, runtime increases linearly with the number of agents, maintaining an almost constant per-agent cost ($\sim$0.56 s), which empirically confirms the $\mathcal{O}(N)$ complexity of MF-LLM.

**Summary.** Both theoretical and empirical analyses demonstrate that MF-LLM scales linearly with population size while preserving simulation fidelity. This scalability enables efficient modeling

Table 7: Scalability of MF-LLM across different population sizes (500–4,000 agents). Lower is better (↓) for all metrics except Micro/Macro-F1 (↑).

| Agents | KL Div. ↓ | Wass. Dist. ↓ | DTW ↓ | Macro F1 ↑ | Micro F1 ↑ | NLL Loss ↓ |
|--------|-----------|---------------|-------|------------|------------|------------|
| 500    | 1.2582    | 0.0645        | 0.1196| 0.3688     | 0.8644     | 3.9916     |
| 1000   | 1.4638    | 0.0729        | 0.1220| 0.3643     | 0.8436     | 4.0112     |
| 2000   | 1.1053    | 0.0685        | 0.0965| 0.3569     | 0.8537     | 4.0236     |
| 3000   | 1.0668    | 0.0650        | 0.0837| 0.3592     | 0.8611     | 4.0142     |
| 4000   | 1.5905    | 0.0698        | 0.1025| 0.3548     | 0.8376     | 4.0237     |

Table 8: Runtime scaling of MF-LLM with respect to agent population size. Results empirically validate the $\mathcal{O}(N)$ complexity.

| Agents | Total Time (s) | Time / Agent (s) |
|--------|----------------|------------------|
| 512    | 280.88         | 0.55             |
| 1008   | 561.89         | 0.56             |
| 1504   | 832.31         | 0.55             |
| 2000   | 1113.41        | 0.56             |
| 2512   | 1404.34        | 0.56             |
| 3008   | 1687.50        | 0.56             |
| 3504   | 1966.29        | 0.56             |
| 4000   | 2249.15        | 0.56             |

of large-scale populations, bridging micro-level decision processes and macro-level population dynamics.

# F  Detailed Experimental Setup and Prompts

## F.1  Dataset

The WEIBO corpus [24] consists of approximately 5,000 real-world events across seven domains: *Crime*, *Culture*, *Health*, *News*, *Politics*, *Sports*, and *Technology*. Each event includes a topic and responses from 100 to 1,000 users, with corresponding user profiles (e.g., location, gender, follower count, activity level) and their posts. We use 4,000 events for training and 1,000 for testing. Training converged after 400 events, at which point optimization was stopped (see Appendix Fig. 6).

## F.2  Baselines.

Based on related work, we summarize the following baseline approaches for modeling other agents' decision-making:

- **State**: The *user profile* and *event topic* serve as the state and input to the LLM, independent of other agents' decisions, as in ElectionSim[44].

- **Recent**: Besides the state, the $k$ most recent actions are provided as input. For instance, TrendSim [45] observes the Top-k comments, and AgentSociety [32] tracks event flow in time order.

- **Popular**: In addition to the state, the $k$ actions with the highest popularity (followers, replies, likes, etc.) are included as input to the LLM, as in OASIS [43] and HiSim[29].

- **SFT**: Standard supervised fine-tuning [30] on state–action pairs for each agent. The baseline is carefully trained to convergence (see training curves in Appendix Fig. 6)

For evaluation, we compare the outputs of our MF-LLM and baselines with ground-truth responses from the WEIBO corpus (hereafter **Real data**).

## F.3 Evaluation Metrics

**Metrics for Evaluating Individual Actions.** To provide a fine-grained assessment of individual agent actions, we define eight evaluation dimensions, each capturing a critical aspect of decision realism. Below, we explain the meaning and motivation behind each dimension.

- **Rumor.** Measures whether the agent is *spreading* or *countering* a rumor. `spread`: The comment believes, forwards, or amplifies the topic discussed. `counter`: The comment challenges, questions, or refutes the truthfulness of the topic, aiming to clarify or correct misinformation.

- **Sentiment.** Captures the emotional tone expressed in the comment, including implicit sarcasm or critique. Categories include `angry`, `calm`, `happy`, `sad`, `fear`, and `surprise`.

- **Attitude.** Reflects the overall emotional orientation of the comment—whether it is *positive*, *negative*, or *neutral*. Particular care is taken to detect subtle negativity even when not explicitly stated.

- **Behavior.** Classifies the type of action taken: `share` if the agent reposts the content, and `comment` if the agent directly evaluates it.

- **Stance.** Identifies the agent's position toward the topic—`support`, `oppose`, or `neutral`. Emphasis is placed on recognizing implicit opposition or dissatisfaction within the comment.

- **Belief.** Assesses whether the agent *believes* in the truthfulness of the discussed topic or expresses *doubt*. Expressions of skepticism, calls for truth, or assertions of falsehood are categorized as `doubt`.

- **Subjectivity.** Distinguishes whether the comment is based on *subjective* personal opinions or on *objective* factual descriptions.

- **Intent.** Captures the underlying purpose of the comment: `question` (asking for clarification), `promotion` (disseminating information), or `opinion` (expressing personal views).

**Prompt Template for Evaluating Decision Dimensions.** We employ `GPT-4o-mini` to evaluate both real and generated action texts. `GPT-4o-mini` offers a highly cost-effective and fast API, enabling stable and consistent evaluation results. We benchmarked several alternatives, including emotion classification models, `GPT-3.5-Turbo`, `GPT-4o`, and `DeepSeek`, and selected `GPT-4o-mini` for its balance between reliability, speed, and deployment cost.

To ensure evaluation reliability, we conducted a human validation study on 100 samples with domain experts (social science scholars, junior faculty, and senior PhD students), comparing human judgments with outputs from multiple LLMs. Table 9 summarizes the results. `GPT-4o-mini` achieves 91.25% agreement with human judgment—higher than `GPT-3.5-Turbo` (79.13%) and comparable to `GPT-4o` (89.38%)—while maintaining the lowest cost and near-real-time speed.

Table 9: Reliability and efficiency comparison of different LLMs for semantic evaluation. `GPT-4o-mini` achieves high human alignment at minimal cost and latency.

| Metric | GPT-4o-mini | GPT-3.5-Turbo | GPT-4o | GPT-4 |
|---|---|---|---|---|
| Agreement w/ Human (%) | 91.25 | 79.13 | 89.38 | **94.13** |
| Price ($/1M tokens, Input / Output) | **0.15/0.60** | 0.5/1.5 | 2.5/10 | 30/60 |
| Speed (s/sample) | 1.25 | **0.83** | 0.96 | 4.02 |

Since semantic evaluation is not a highly complex task for modern LLMs, `GPT-4o-mini` provides the most cost-effective balance of accuracy, efficiency, and scalability. Below, we present the prompt template used for decision-dimension evaluation.

## Prompt Template for Evaluating Decision Dimensions

**Role:** You are an expert in public opinion content analysis. **Task:** Analyze multiple user comments objectively to evaluate the sentiment, stance, and opinion orientation regarding the following topic:
**Discussion Topic:** {topic}

**Instructions:** For each comment, perform analysis according to the following nine dimensions and strictly return the results in `JSON` format. **Special Note:** The emoji "@_@" typically conveys feelings of "surprise, confusion, or being stunned".

1. **rumor** (Rumor propagation): Select from `["spread", "counter"]`.
   - `"counter"`: Comments that aim to refute, disbelieve, question the authenticity, expect further clarification, or directly point out the falsity of the topic.
   - `"spread"`: All other comments, including those expressing belief in the topic, reposting content, tagging usernames, repeating topic content, or expressing emotional reactions to the topic.

2. **sentiment** (Emotional state): Capture the user's emotional state conveyed through the comment (including punctuation and tone). Choose from `["angry", "calm", "happy", "sad", "fear", "surprise"]`. *Note:* Simple reposts are categorized as `"calm"`.

3. **attitude** (Attitude polarity): Determine whether the user's sentiment is positive, negative, or neutral. Choose from `["positive", "negative", "neutral"]`. *Note:* Pay close attention to any implicit negative sentiment (e.g., sarcasm, criticism).

4. **behavior** (Behavior type): Select from `["comment", "share"]`.
   - `"share"`: The comment is primarily forwarding or reposting content.
   - `"comment"`: The comment expresses an evaluation, opinion, or reaction.

5. **stance** (Stance towards the topic): Select from `["support", "oppose", "neutral"]`. *Note:* Pay attention to implicit opposition, dissatisfaction, or criticism.

6. **belief** (Belief in the topic): Select from `["believe", "doubt"]`.
   - `"believe"`: The comment expresses belief in the topic (including reposting).
   - `"doubt"`: The comment questions, refutes, or expresses skepticism towards the topic.

7. **keywords** (Keyword extraction): Extract important keywords from the comment and return them as an array (e.g., `["policy", "economy"]`). If the comment is meaningless, return `[""]`.

8. **subjectivity** (Subjectivity): Determine whether the comment is based on subjective opinions or objective facts. Select from `["subjective", "objective"]`.

9. **intent** (Intent classification): Select from `["question", "promotion", "opinion"]`.
   - `"question"`: The comment primarily asks a question.
   - `"promotion"`: The comment primarily disseminates or promotes information.
   - `"opinion"`: The comment primarily expresses an opinion or viewpoint.

**Input Format:** The following are {len(comments)} user comments:

```
Comment 1: "{comment_1}"
Comment 2: "{comment_2}"
...
```

**Output Format:** Strictly return a JSON array evaluating the {len(comments)} comments. Do not output anything other than the JSON array.

**Example Output:**

```
[
  {
    "rumor": "spread",
    "sentiment_state": "calm",
    "sentiment_tendency": "neutral",
    "behavior_type": "share",
    "stance": "neutral",
    "belief_degree": "believe",
    "keywords": ["share", "weibo"],
```

```
    "subjectivity": "objective",
    "intent_classification": "promotion"
  },
  {
    "rumor": "counter",
    "sentiment_state": "angry",
    "sentiment_tendency": "negative",
    "behavior_type": "comment",
    "stance": "oppose",
    "belief_degree": "doubt",
    "keywords": ["fake", "impossible"],
    "subjectivity": "subjective",
    "intent_classification": "opinion"
  }
]
```

## F.4 Prompt Templates

In this paper, we design three prompt templates. (i) The first template guides the policy model to generate individual decision-making behaviors. (ii) The second template assists the mean field model in summarizing the evolving population distribution. (iii) The third template enables `GPT-4o-mini` to evaluate the dimensions of individual decisions with fine-grained analysis. Together, these templates ensure a coherent workflow across decision generation, mean field updating, and evaluation.

---

**Prompt Template for Policy Model**

You are tasked with simulating a plausible user action (reposting or commenting) based on their profile and the current population information.

**Inputs:**

- **Discussion Topic**: {topic}
- **Recent Comments (if available)**: {Recent_comment}
- **Popular Comments (if available)**: {Popular_comment}
- **Current Mean Field**: {mean_field}
- **User Profile Attributes**:
  - **Location**
  - **Description**
  - **Gender**: Male or Female
  - **Number of Friends**: Categorized as
    * Very few: fewer than 10
    * Few: 10 to 30
    * Moderate: 31 to 1000
    * Many: 1001 to 3000
    * Very many: more than 3000
  - **Number of Followers (Influence Level)**: Categorized as
    * Very low: fewer than 100
    * Low: 101 to 500
    * Moderate: 501 to 1000
    * High: 1001 to 10000
    * Very high: more than 10000
  - **Activity Level (Based on Total Interactions)**:
    * Inactive: fewer than 10 interactions
    * Moderately active: 10 to 100 interactions
    * Highly active: more than 100 interactions
  - **Verification Status**:
    * Verified user (with verification type ID)
    * Non-verified user

---

> **Intermediate Step (User State Construction):** First, generate a concise user profile description in Chinese (or English), following the template:
>
> > A user from {user_location}, described as {user_description}, identified as {gender}, with a {friends_level} number of friends and a {influence_level} level of influence based on followers. The user is {activity_level} in terms of interactions, and the account is {verified_status}.
>
> **Note:** If using GPT or DeepSeek models, we add an explicit instruction at the end of the prompt: *"Output only the final simulated text without any intermediate reasoning process."* In the output generated by `DeepSeek-R1 Distill-Qwen-32B`, any content enclosed within `<think>...</think>` tags is removed, and only the final generated text is preserved for evaluation.

---

### Prompt Template for Mean Field Model

You are tasked with summarizing the distribution of user comments regarding a specific discussion topic.

**Inputs:**

- **Discussion Topic**: {topic}
- **Previous Mean Field**: {previous_mean_field}
- **Recent User Comments**: A list of user comments formatted as "Comment 1: xxx", "Comment 2: xxx", etc.

**Instructions:** Based on the provided information, summarize the overall user discussion by addressing the following six aspects in order of importance:

1. **Stance Distribution**: Are users predominantly supportive or oppositional?
2. **Opinion Distribution**: What are the major viewpoints expressed?
3. **Emotion Distribution**: Are emotions primarily anger, excitement, doubt, or anxiety? Overall, are sentiments positive, negative, or neutral?
4. **Behavior Distribution**: Are users more inclined to repost or to comment?
5. **Perception of Topic Authenticity**: To what extent do users believe or doubt the authenticity of the topic?
6. **Intent of Comments**: Are users primarily asking questions, expressing opinions, or disseminating information?

**Response Requirements:** Provide a concise summary in Chinese (or English), approximately 200 words in length. Ensure the response is structured clearly, focuses on key points, and remains easy to comprehend.

---

## G  Details of MF-LLM Framework

### G.1  Derivation of the IB-Tune Objective

**Variational upper-bound on the $I(m_t; X)$ term.** We begin from the definition of mutual information and derive a computable bound by introducing a variational prior $r(m_t)$.

$$
\begin{aligned}
I(m_t; X) &= \mathbb{E}_{p(X, m_t)}\left[\log \frac{p(m_t \mid X)}{p(m_t)}\right] && \text{(definition of mutual information)} \\
&= \mathbb{E}_{p(X)}\left[\text{KL}\big(p(m_t \mid X) \,\|\, p(m_t)\big)\right] && \text{(rewriting as an expectation of KL)} \\
&\approx \mathbb{E}_{p(X)}\left[\text{KL}\big(\mu_\phi(m_t \mid X) \,\|\, p(m_t)\big)\right] && \text{(variational posterior } \mu_\phi) \\
&= \mathbb{E}_{p(X)}\mathbb{E}_{\mu_\phi(m_t \mid X)}\left[\log \mu_\phi(m_t \mid X) - \log p(m_t)\right].
\end{aligned}
$$

Next, we add and subtract the log-density of a fixed prior $r(m_t)$:

$$
\log \mu_\phi - \log p = (\log \mu_\phi - \log r) + (\log r - \log p).
$$

Hence
$$I(m_t; X) = \underbrace{\mathbb{E}_{p(X)}\mathrm{KL}\big(\mu_\phi(m_t \mid X) \,\|\, r(m_t)\big)}_{(A)} + \underbrace{\mathbb{E}_{p(X)}\mathbb{E}_{\mu_\phi(m_t\mid X)}\big[\log r(m_t) - \log p(m_t)\big]}_{(B)}.$$

Observe that
$$(B) = \mathbb{E}_{p(m_t)}\big[\log r(m_t) - \log p(m_t)\big] = -\mathrm{KL}\big(p(m_t) \,\|\, r(m_t)\big) \le 0.$$

Discarding this non-negative term yields the desired bound:
$$I(m_t; X) \le \mathbb{E}_{p(X)}\mathrm{KL}\big(\mu_\phi(m_t \mid X) \,\|\, r(m_t)\big) = \mathbb{E}_{p(X)}\mathbb{E}_{\mu_\phi(m_t\mid X)}\big[\log \mu_\phi(m_t \mid X) - \log r(m_t)\big].$$

### G.2   Scaling to More Agents via State Sampling

**Simulating More Agents**   Our MF-LLM framework is inherently scalable and can simulate an arbitrary number of agents over time through iterative rollouts of the policy model and mean field model, as shown in Figure 2. A key question is how to obtain the state information for additional agents during simulation. If the profiles of future users or agents are available, they can be directly used as the input states for simulation. However, *in scenarios where the profiles of future agents are unknown*—such as when predicting future participation—**we extend our method by sampling agent states from an existing state distribution.** Agent profile distributions tend to evolve much more slowly than opinion/decision distributions, especially relative to the fast dynamics of event-driven discussions. Therefore, for long-term simulations of collective decision trends, it is reasonable to sample agent states from historical user profiles. Our experimental results further suggest that the mean field model plays a dominant role in shaping collective dynamics, while the specific agent state distribution has a relatively minor impact.

## H   Full Version of Related Work

### H.1   Agent-Based Models: Foundations and Limitations.

Agent-Based Modeling (ABM) has long served as a cornerstone for simulating complex social systems, enabling emergent phenomena to arise from local interactions among individuals. Seminal works such as Sugarscape [10], Bonabeau's survey [6], and domain-specific applications in crowd dynamics [15], market simulations [36], ecosystems [14], and public policy [2, 26] have demonstrated the versatility of ABMs across diverse domains. However, traditional ABMs often rely on manually crafted behavioral rules, limiting their adaptability and scalability—especially as population size and environmental complexity increase [7, 17, 25]. While effective at capturing rule-driven emergent patterns, ABMs typically lack the flexibility required to generalize across diverse and unseen scenarios. Recent advances in large language models (LLMs) offer a promising direction to overcome these limitations by equipping agents with generative decision-making capabilities grounded in rich contextual knowledge.

### H.2   LLMs for Social Simulation: Progress and Gaps.

The rise of LLMs has catalyzed a new wave of social simulation systems, enabling agents to reason, adapt, and interact through natural language. Early works such as Generative Agents [31] and Sotopia [48] demonstrated the potential of LLMs to support small-scale, open-ended simulations. Recent efforts have scaled up to broader domains, including social media [43, 45, 32], elections [44], economic behavior [22, 42], and misinformation spread [23]. Toolkits such as GenSim [35], AgentScope [13], and AgentSociety [32] have facilitated the construction of complex multi-agent environments. Despite these advances, key limitations remain. Many simulations rely on handcrafted prompts, fixed roles, and heuristic memory mechanisms that lack alignment with real-world behavior, limiting their quantitative fidelity. Agent interactions are typically governed by static or scripted schedules [43, 45, 8], preventing agents from adapting to changing collective dynamics. Although recent approaches attempt to incorporate peer behavior and scene evolution into memory [23, 45, 29, 43], they often depend on heuristic retrieval or summarization strategies, lacking principled mechanisms to retain decision-critical information over long horizons. These constraints hinder the simulation of realistic, adaptive population behavior. Progress in LLM-based social simulation will require principled mechanisms to model the feedback loop between individual actions and evolving population signals—allowing agents to both respond to and influence macro-level dynamics.

### H.3 Mean Field Approximation: Scalable Interaction Modeling.

Mean field approximation offers an effective way to model large multi-agent systems by replacing costly pairwise interactions with interactions between each agent and a shared population signal [19, 16, 41]. This abstraction is formalized in Mean Field Game (MFG) theory, which enables scalable modeling of individual decisions and their aggregate influence on population dynamics. By collapsing multi-agent dependencies into a compact population-level representation, MFGs significantly reduce the complexity of interaction modeling and have been applied to domains such as social influence [40, 4], traffic control [11, 39], energy optimization [9], economic policy [27], and imitation learning [46]. However, classical MFGs often assume stylized agent behaviors and lack the contextual reasoning required for realistic social simulation. Recent neural variants [20, 21] improve expressiveness, but remain less flexible than large language models. Motivated by these limitations, we propose **Mean-Field LLM (MF-LLM)**, which combines the scalability of mean-field approximation with the generative reasoning capabilities of LLMs. MF-LLM explicitly models the bidirectional feedback between individual behavior and evolving population dynamics, enabling scalable, high-fidelity social simulation.

## I  Showcase of LLM-Generated Content

In Appendix I, we present a subset of generated comments under the "Liu Xiang Olympic Rumor" scenario, corresponding to the decision distribution at a single time step (manifested as a comments distribution in this context). We compare outputs generated by agents in identical states but using different methods and LLM backbones. All comments were initially generated in Chinese and faithfully translated into English for presentation. The complete results are available in our GitHub repository.

### I.1  Analysis of Generated Comments across Different Algorithms

To further evaluate the fidelity of generated comments, we conducted a comparative analysis along three key dimensions: content richness and naturalness, intention distribution (rumor-spreading vs. countering vs. neutral), and belief uncertainty. We compare the outputs from *MF-LLM* and *State* generation against real human comments collected from the same context.

**Content Richness and Naturalness.**  Comments generated by MF-LLM exhibit a high degree of linguistic diversity and conversational naturalness, closely resembling real human discourse. Examples include casual expressions such as "Haha  Is this news real?" and mixed emotional cues like "This feels way too real... but let's wait for the official announcement." These stylistic features mirror the informal, fragmented, and emotion-driven nature of real-world social media discussions. In contrast, *State* generation tends to produce more formal, structured outputs, often resembling official clarifications (e.g., "Regarding the latest news, I have contacted the relevant parties, please do not believe rumors."). Such responses, while coherent, lack the spontaneous variability observed in human comments.

**Intention Distribution.**  MF-LLM successfully captures the coexistence of rumor-spreading, rumor-debunking, and neutral stances within the population. Some generated comments actively propagate suspicions (e.g., "Stop joking around."), while others explicitly advise caution against misinformation (e.g., "Fake people and fake stories are all rumors, don't believe them!"). Importantly, MF-LLM also generates comments that reflect passive spectatorship, mimicking users who merely observe without taking a stance. In contrast, *State* generation is skewed toward debunking or formal clarification, showing limited presence of rumor-spreading or ambivalent behaviors. This mismatch leads to a narrower and less realistic distribution compared to real-world discussions.

**Belief Uncertainty.**  Real human comments display a wide spectrum of belief, ranging from full acceptance to skepticism and outright denial. MF-LLM-generated comments faithfully reproduce this belief heterogeneity. Some comments show partial belief or hesitation (e.g., "Is it true? Or fake? Too many rumors going around."), while others demonstrate firm rejection or concern. This spread reflects the nuanced and non-binary attitudes common in organic social discourse. Conversely, *State* generation primarily produces comments that firmly oppose rumor acceptance, lacking examples that inhabit the uncertain middle ground.

**Summary.** Overall, MF-LLM significantly outperforms *State* generation in replicating the natural distribution of real human comments across multiple dimensions. It captures richer language patterns, more balanced intention distributions, and a realistic belief spectrum, enabling more faithful simulation of public opinion dynamics.

## I.2 Analysis of Repetition across Different LLMs

We further evaluate the quality of generated comments by examining content diversity, alignment with real-world distributions, and model-specific generation tendencies. In particular, we compare `GPT-4o-mini` and `DeepSeek-R1 Distill-Qwen-32B` against `Qwen2-1.5B-Instruct MF-LLM/State`.

**High Repetition Across Agent States.** Both `GPT-4o-mini` and `DeepSeek-R1` exhibit significant repetition across outputs, even when agent states vary. For example, `GPT-4o-mini` frequently repeats phrases like "really heartbreaking" and "sigh," while `DeepSeek-R1` consistently generates sensational headlines such as "shocking! Liu Xiang's withdrawal scandal exposed!" across different user profiles. We highlight repeated content using color in following tables. This redundancy indicates a failure to adapt responses to agent-specific contexts, limiting the models' ability to simulate diverse population behavior.

**Deviation from Real Comment Distributions.** Generated comments from `GPT-4o-mini` and `DeepSeek-R1` deviate markedly from real-world comment patterns. Human comments naturally span rumor-spreading, debunking, passive observation, humor, and emotional venting. In contrast, both models collapse this diversity into narrow emotional templates, predominantly emphasizing grief or outrage. This leads to unrealistic collective behavior, missing the rich spectrum of opinions and intentions present in authentic discussions.

**Distinct Narrative Biases.** Each model also imposes distinctive stylistic biases. `GPT-4o-mini` favors sentimental and empathetic narratives, often regardless of the agent's initial belief state. `DeepSeek-R1`, in contrast, exaggerates sensationalism by systematically inserting attention-grabbing phrases. These narrative tendencies further distort the natural variance of social discourse, amplifying specific emotional tones beyond their prevalence in *Real data*.

**Summary.** Compared to `Qwen2-1.5B-Instruct MF-LLM`, both `GPT-4o-mini` and `DeepSeek-R1` exhibit higher repetition, stronger narrative bias, and weaker alignment with real-world collective decision-making dynamics. These results highlight the necessity of diversity-aware generation strategies when modeling large-scale social behaviors.

**Comparison between Real and Generated Comments (Qwen2-1.5B-Instruct MF-LLM)**

| Index | Real Comment | Generated Comment |
|---|---|---|
| 1 | Repost. | I feel the same way!! But let's wait for the official announcement. |
| 2 | Stop spreading bullshit rumors, you brainless nationalists! [angry] Since when could the host country just change the IAAF competition rules whenever they want?! [angry] How much more stupid can you get just to suck up to someone?! | Haha~ Is this news real? Were we tricked again? |
| 3 | Is it true? Or fake? Too many rumors going around. | @RealNewsBot Fake people and fake stories are all rumors, don't believe them! |
| 4 | Repost. | This feels way too real. @CRNews @SohuVideo This whole thing is really confusing... //@CRNews: Didn't expect this~ //@SohuVideo: [haha] That was close [thinking][sly smile] |
| 5 | [Watching] | Stop joking around. Only China would dare to pull something like this. Heh. |
| 6 | Is it real or fake? | You know what's up. Stop spreading rumors, haha, seriously. |
| 7 | No way, really? Is it true? @lovelingzi | @RealWorld @OnlyBelieveInSkills @SportsNews |
| 8 | [angry] Shameful. | @BeijingSportsChannel |
| 9 | Looking for the truth. | Repost. |
| 10 | Repost. | "Rumor says ________" |
| 11 | Is this really how it happened?! [angry] | @Everyone Please pay attention~ Recently a shocking piece of news has been spreading online...[image]... Let's check out this video together... |
| 12 | Repost. | Why are there so many people spreading fake news? Stop lying!! Is all of this even true? I was still hoping someone would tell the truth~ |
| 13 | Damn it! Inhumane! I'm boycotting British goods! [angry rant] | @TruthBot @ChinaTrackTeam I'm really curious why this issue is causing such a huge debate? We need to see if there's any solid evidence proving this claim, because online rumors are everywhere now... //@BeijingSports: Very... |
| 14 | Repost. | @People's Daily: |
| 15 | Paranoia kicking in again. | @Hello everyone~ Here's the latest trending news~ As we all know, there was a shocking report earlier saying that China's track star Liu Xiang might have messed up because the hurdle spacing was deliberately altered. |

## Comparison between Real and Generated Comments (Qwen2-1.5B-Instruct State)

| Index | Real Comment | Generated Comment |
|---|---|---|
| 1 | Repost. | I saw someone post a message about Liu Xiang withdrawing before the London Olympics. He said it was because a hurdle spacing was lengthened... |
| 2 | Stop spreading bullshit rumors, you brainless nationalists! [angry] Since when could the host country just change the IAAF competition rules... | [] |
| 3 | Is it true? Or fake? Too many rumors going around. | [Regarding the latest news on this topic, I have already contacted the relevant parties, please everyone don't believe in rumor spreading ] |
| 4 | Repost. | Shocked!! It's spreading wildly online that Liu Xiang was accused of violating IAAF rules, and that improper setup of the competition venue... |
| 5 | [Watching] | Really? Does this maybe reveal some of our own problems? |
| 6 | Is it real or fake? | [Shocking rumor online: Liu Xiang's withdrawal involved a huge scandal? The hurdle spacing was actually widened...] |
| 7 | No way, really? Is it true? @lovelingzi | @RealWorld @MeToo |
| 8 | [angry] Shameful. | [Shocking rumor online: Liu Xiang's withdrawal stunning inside story — London's carelessness actually led to hurdle spacing increase!] |
| 9 | Looking for the truth. | #OlympicQualifiers##ShockingSecret After Liu Xiang's hurdle misstep, how did he still manage to break through?... |
| 10 | Repost. | "Rumor says ______" |
| 11 | Is this really how it happened?! [angry] | [Reply] @Anonymous: This topic is very interesting. I think the key problem lies in differences in the venue and training levels... |
| 12 | Repost. | "Haha......" Netizen comment: Haha...... |
| 13 | Damn it! Inhumane! I'm boycotting British goods! [angry rant] | @Liu Xiang questioned for continuous hurdle mistakes — is this claim credible? |
| 14 | Repost. | @People's Daily: |
| 15 | Paranoia kicking in again. | #Today'sTopic #OnlineRumorEvent |

## Comparison between Real and Generated Comments (GPT-4o-mini State)

| Index | Real Comment | Generated Comment |
|---|---|---|
| 1 | Repost. | Although Liu Xiang's withdrawal is really regrettable, this kind of hurdle spacing change is just way too unprofessional! He made so many adjustments... |
| 2 | Stop spreading bullshit rumors, you brainless nationalists! [angry] Since when could the host country just change the IAAF competition rules whenever they want?! [angry] How much more stupid can you get just to suck up to someone?! | Sigh, Liu Xiang's withdrawal this time is really heartbreaking! I heard the hurdle spacing was actually changed, this is just way too unreasonable!... |
| 3 | Is it true? Or fake? Too many rumors going around. | Liu Xiang's withdrawal is really heartbreaking, especially with the hurdle spacing problem in London, it really should not have happened!... |
| 4 | Repost. | After seeing this behind-the-scenes story, I truly feel a mix of emotions. Liu Xiang, as an excellent athlete, has been through so much... |
| 5 | [Watching] | I heard Liu Xiang's withdrawal in London was because of the hurdle spacing change, this is just too outrageous!... |
| 6 | Is it real or fake? | Liu Xiang's withdrawal is really heartbreaking, seeing such behind-the-scenes news makes me feel so emotional... |
| 7 | No way, really? Is it true? @lovelingzi | Sigh, this news is really heartbreaking! Liu Xiang has always been a hero in our hearts, and his seven-step technique... |
| 8 | [angry] Shameful. | As a fan who has been following Liu Xiang, I am truly shocked by the behind-the-scenes of this London event!... |
| 9 | Looking for the truth. | Sigh, Liu Xiang's withdrawal this time is really heartbreaking! I have always supported him, and I never thought that... |
| 10 | Repost. | Seeing Liu Xiang at the London Games was really heartbreaking! We originally thought he would perform well, but unexpectedly... |
| 11 | Is this really how it happened?! [angry] | This is really regrettable! Liu Xiang has always been our pride, but unexpectedly in London, because of the careless hurdle spacing design... |
| 12 | Repost. | In this shocking piece of news, I feel extremely regretful and disappointed. Liu Xiang's seven-step technique was a classic skill... |
| 13 | Damn it! Inhumane! I'm boycotting British goods! [angry rant] | The inside story of Liu Xiang's withdrawal in London is truly shocking! It turns out the hurdle spacing was actually increased to 13.85 meters... |
| 14 | Repost. | Seeing the inside story about Liu Xiang's withdrawal, my heart really feels full of mixed emotions. Turns out, the hurdle spacing change... |
| 15 | Paranoia kicking in again. | Sigh, it's really too disappointing! Behind Liu Xiang's withdrawal, there was actually such a hidden story... |

## Comparison between Real and Generated Comments (DeepSeek-R1 Distill-Qwen-32B State)

| Index | Real Comment | Generated Comment |
|---|---|---|
| 1 | Repost. | Repost Weibo: Shocking! The real reason behind Liu Xiang's withdrawal scandal exposed! Carelessness at the London Olympics led to an increase in hurdle spacing... |
| 2 | Stop spreading bullshit rumors, you brainless nationalists! [angry] ... | "The hurdle spacing increased by 0.13 meters, it looks like a tiny change, but it had a huge impact on Liu Xiang's seven-step technique! This small detail at the London Olympics is really chilling..." |
| 3 | Is it true? Or fake? Too many rumors going around. | Liu Xiang is really too unfortunate, the hurdle spacing adjustment affected his performance, and other athletes also had problems. This is so unfair! Hope the truth will be revealed! |
| 4 | Repost. | Shocked! London really screwed Liu Xiang this time! The hurdle spacing increased by 0.13 meters... |
| 5 | [Watching] | Shocking! Liu Xiang's withdrawal scandal exposed! The hurdle spacing at the London Olympics was actually increased to 13.85 meters... |
| 6 | Is it real or fake? | Liu Xiang's withdrawal is indeed regrettable, but the hurdle spacing change at the London Olympics was definitely a key factor... |
| 7 | No way, really? Is it true? @lovelingzi | Shocking! Liu Xiang's withdrawal scandal exposed! Carelessness at the London Olympics actually lengthened the hurdle spacing... |
| 8 | [angry] Shameful. | [Repost]Liu Xiang's withdrawal scandal exposed! London Olympics adjusted the hurdle spacing, from 13.72 to 13.85... |
| 9 | Looking for the truth. | This is so unfair! Liu Xiang's seven-step technique was clearly designed for 13.72 meters... |
| 10 | Repost. | This is way too unfair! Liu Xiang's seven-step technique was designed for a 13.72-meter distance, but London actually changed it... |
| 11 | Is this really how it happened?! [angry] | Netizen emotions: sadness, disappointment, support for Liu Xiang. |
| 12 | Repost. | Shocking! There's such a big scandal behind Liu Xiang's withdrawal! Was the hurdle spacing at the London Olympics really set so casually?... |
| 13 | Damn it! Inhumane! I'm boycotting British goods! [angry rant] | This is way too outrageous! The London Olympics actually changed the hurdle spacing, causing Liu Xiang to hit the hurdle directly... |
| 14 | Repost. | Oh my god, Liu Xiang's withdrawal is really too unfortunate! London increased the hurdle spacing to 13.85 meters... |
| 15 | Paranoia kicking in again. | Sigh, this is way too outrageous! Liu Xiang's seven-step technique was his signature move... |

