# OpenReview forum: "MF-LLM: Simulating Population Decision Dynamics via a Mean-Field Large Language Model Framework"
_NeurIPS.cc/2025/Conference — NeurIPS 2025 poster_

### Official Review · Reviewer_G3Ud · 2025-06-30

**Clarity:** 2
**Significance:** 3
**Originality:** 3
**Rating:** 4
**Confidence:** 3

**Summary:**

This paper introduces MF-LLM, the first framework to integrate mean-field theory into LLM-based social simulation by iteratively exchanging population-level signals and individual agent decisions to capture coherent collective behavior. To improve alignment with real-world data, it proposes IB-Tune, an Information Bottleneck–inspired fine-tuning method that compresses population history into only those signals most predictive of future actions. Evaluated on a Weibo dataset, MF-LLM reduces KL divergence to human population distributions by 47% over non-mean-field baselines, generalizes across seven domains and four LLM backbones, and supports accurate trend forecasting and intervention planning.

**Questions:**

See Weaknesses.

**Ethical Concerns:**

["NO or VERY MINOR ethics concerns only"]

**Final Justification:**

The rebuttal is convincing, and I have adjusted my score.

**Limitations:**

See Weaknesses.

**Quality:**

2

**Strengths And Weaknesses:**

Strengths:
-	This paper introduces a novel MF-LLM framework that seamlessly integrates mean-field theory with LLM-based social simulation to capture bidirectional dynamics between population signals and individual decisions.
-	Extensive experiments on a large Weibo dataset, across seven domains and four LLM backbones, demonstrate robust performance improvements and scalability.
-	A comprehensive analysis—including KL divergence comparisons, trend-forecasting results, and intervention-planning charts—provides clear, interpretable evidence of the framework’s effectiveness.

Weaknesses:
-	The paper claims a general-purpose social simulation framework but is evaluated on only a single Chinese Weibo dataset. As a general method, it should be tested on at least 2–3 diverse datasets. Besides, include at least one English dataset.
-	Scalability is cited as a main motivation, yet the paper lacks any complexity analysis or detailed experiments on efficiency and scalability.
-	The writing should be improved. For example, the contributions paragraph is overly lengthy and fails to highlight the core innovations succinctly.

---

> ### Author Rebuttal · Authors · 2025-07-31
>
> **Dear Reviewer G3Ud,**
>
> Thank you very much for taking the time to review MF‑LLM and for your valuable feedback, as well as for recognizing our work with *“extensive experiments,” “robust performance improvements,” and “comprehensive analysis providing clear, interpretable evidence of the framework’s effectiveness.”*
>
> During the rebuttal, we have made every effort to address your concerns through additional experiments, detailed analysis, and writing refinements:
>
> - **Added experiments on diverse datasets** — including English Twitter replies and Bitcoin sentiment — showing that MF‑LLM generalizes across languages and diverse forms of collective behavior.
> - **Provided scalability analysis** — formally analyzing complexity reduction from O(N²) to O(N) and demonstrating efficiency gains in large‑scale simulation.
> - **Extended long‑horizon evaluation** — adding experiments up to 2,000 steps to confirm MF‑LLM’s robust performance at scale.
> - **Refined the contribution summary** — presenting the core innovations more concisely to directly address the reviewer’s clarity concerns.
>
> **Below, we provide detailed responses to each of your questions, with the utmost sincerity, to address every concern.**
>
>
>
> ---
>
>
>
> > **W1: "The paper claims a general-purpose social simulation framework but is evaluated on only a single Chinese Weibo dataset. As a general method, it should be tested on at least 2–3 diverse datasets. Besides, include at least one English dataset."**
>
> **Answer:** Thank you for this important question. MF-LLM is inherently culture-agnostic, with no assumptions about language or cultural background. To validate its generalization ability, we added experiments on **culturally diverse English datasets** without domain-specific fine-tuning.
>
> **(1) Famous Keyword Twitter Replies Dataset** (Kaggle)
>
>  This dataset covers globally relevant topics (COVID, Vaccine, NFT, TikTok, WorldCup, etc.). Using Qwen2-1.5B-Instruct, MF-LLM consistently outperforms baselines:
>
> | Method     | KL↓        | WD↓        | DTW↓       | NLL↓       | Macro F1↑  | Micro F1↑  |
> | ---------- | ---------- | ---------- | ---------- | ---------- | ---------- | ---------- |
> | State      | 0.8696     | **0.0673** | 0.1700     | 4.2121     | 0.4977     | **0.8241** |
> | Recent     | 1.5061     | 0.0922     | 0.2206     | 4.1053     | 0.4471     | 0.7661     |
> | Popular    | 1.0059     | 0.0725     | 0.1553     | 4.0697     | 0.4864     | 0.8047     |
> | **MF-LLM** | **0.6883** | 0.0744     | **0.1587** | **3.9748** | **0.4989** | 0.8093     |
>
> **(2) Market Behavior Simulation**
>
>  To assess MF-LLM’s generalization to market behavior, we refer to *TwinMarket* [1], which uses real-world transaction data from Chinese social platforms (Xueqiu, Guba), and *ElectionSim* [2], which uses Twitter data on elections. However, neither releases datasets, making direct comparison infeasible.
>
> Following their approach of using social media data to simulate collective decisions, we use Bitcoin-related tweet–reply data from Twitter to simulate dynamic **bullish/bearish sentiment** at the population level. Once again, MF-LLM achieves superior alignment:
>
> | Method     | KL↓        | WD↓        | DTW↓       | NLL↓       | Macro F1↑  | Micro F1↑  |
> | ---------- | ---------- | ---------- | ---------- | ---------- | ---------- | ---------- |
> | State      | 1.2304     | 0.0854     | 0.4505     | 4.4320     | 0.4723     | 0.6721     |
> | Recent     | 2.0641     | **0.0800** | 0.4730     | 4.3438     | 0.4550     | 0.6573     |
> | Popular    | 1.4394     | 0.0853     | 0.3930     | 4.2537     | 0.4834     | 0.6920     |
> | **MF-LLM** | **0.9316** | 0.0915     | **0.2364** | **4.1485** | **0.5025** | **0.7741** |
>
>
>
> **We hope these additional results address your concern by showing that MF-LLM generalizes across languages and cultural settings.** These findings will be incorporated into the revised paper.
>
>
>
> [1] TwinMarket: A Scalable Behavioral and Social Simulation for Financial Markets.
>
> [2] ElectionSim: Massive Population Election Simulation Powered by LLM Agents.
>
> ---
>
>
>
> > **W2: "Scalability is cited as a main motivation, yet the paper lacks any complexity analysis or detailed experiments on efficiency and scalability."**
>
> **Answer:**
>
> **(1) Computational Complexity Analysis**:
>
> MF-LLM aims to capture dynamic agent interactions, which are fundamental to the formation of collective behavior. **Modeling all agent–agent pairwise interactions requires O(N²) complexity, while MF-LLM reduces this to O(N) linear complexity.** Specifically, the policy model generates one personalized action per agent, leading to O(N) LLM inferences. The mean field model updates the population signal once per batch of K agents, resulting in O(N/K) mean field inferences.
>
> Overall, MF-LLM reduces total LLM inference complexity to O(N) while still modeling agent interactions and producing population-level dynamics. This contrasts with baselines that either ignore agent interactions entirely or approximate them through manual prompt engineering without dynamic modeling.
>
> **(2) Testing on Large Populations**:
>
> In the original paper, the Weibo dataset naturally limited simulation horizons to ~900 steps due to the discussion length of real events. To further validate scalability, we extend evaluation to the English Twitter dataset with long-horizon discussions (up to 2,000 steps). **Results show that while error increases slightly with horizon, MF-LLM consistently outperforms baselines.**
>
>
> | Method     | KL↓        | WD↓        | DTW↓       | NLL↓       | Macro F1↑  | Micro F1↑  |
> | ---------- | ---------- | ---------- | ---------- | ---------- | ---------- | ---------- |
> | State      | 1.1701     | 0.0714     | 0.1110     | 4.2050     | **0.3773** | 0.8456     |
> | Recent     | 1.8701     | 0.0685     | 0.1178     | 4.1000     | 0.3210     | 0.8533     |
> | Popular    | 1.4291     | 0.0722     | **0.0937** | 4.0863     | 0.3475     | 0.8438     |
> | **MF-LLM** | **1.1053** | **0.0685** | 0.0965     | **4.0236** | 0.3569     | **0.8537** |
>
> ---
>
>
>
> > **W3: "The writing should be improved. For example, the contributions paragraph is overly lengthy and fails to highlight the core innovations succinctly."**
>
> **Answer:** We thank the reviewer for the valuable suggestion. **We will further polish the writing in the revised paper**. Below is a more concise summary of the contributions, highlighting the core innovations:
>
> **(1) MF‑LLM framework** — We introduce MF‑LLM, the first framework to integrate mean‑field theory into LLM‑based social simulation, by iteratively updating population signals and individual decisions to capture the evolving trajectories of collective behavior.
>
> **(2) IB‑Tune alignment** — To improve alignment with real‑world data, we introduce IB‑Tune, an Information Bottleneck–inspired fine‑tuning method that compresses population history into only those population signals most predictive of future actions.
>
> **(3) Empirical validation** — MF‑LLM achieves up to **47% lower KL divergence** than baselines, generalizes across **seven domains** and **four LLM backbones**, and supports real‑world forecasting and intervention planning.
>
> ---
>
>
>
> **We hope these clarifications and new results will encourage you to reconsider your score. If you have any further questions or would like additional clarifications, we would be very happy to discuss them.**

---

> ### Author Response · Authors · 2025-08-04
> **Hope to Get Your Reply**
>
> **Dear Reviewer G3Ud,**
>
> Once again, we sincerely thank you for the time and effort you have dedicated to reviewing our paper. As the discussion period is approaching its close, we are eager to receive your feedback on our response. We fully understand your busy schedule, but we would greatly appreciate it if you could take our response into account when updating your rating and discussing with the AC and other reviewers.
>
> In addition to the experiments **already presented in the rebuttal**, including:
>
> - Experiments on English datasets (Twitter) and market behavior simulation,
> - Simulation experiments extending to 2000 steps,
> - Computational Complexity Analysis,
>
> we have **added further experiments** to strengthen our claims:
>
> ------
> > **W1: "The paper claims a general-purpose social simulation framework but is evaluated on only a single Chinese Weibo dataset. As a general method, it should be tested on at least 2–3 diverse datasets. Besides, include at least one English dataset."**
>
> We **conducted additional evaluations** of MF‑LLM (without fine‑tuning) on the **Twitter dataset** using `GPT‑4o‑mini`, `GPT‑4o`, and `DeepSeek‑R1‑Distill‑Qwen‑32B` backbones. Results show that MF‑LLM consistently generalizes across cultural contexts.
>
> | Method | KL↓        | WD↓        | DTW↓       | Macro F1↑  | Micro F1↑  |
> | - | - | - | - | - | - |
> | `GPT‑4o‑mini`    |   |   |   |   |   |
> | State  | 6.8382     | **0.1469** | 0.4513     | 0.2808     | 0.6282     |
> | Recent | 6.5273     | 0.1491     | 0.4444     | 0.2843     | 0.6361     |
> | Popular| 7.5150     | 0.1634     | 0.4251     | 0.2526     | 0.6112     |
> | **MF‑LLM** | **5.9322** | 0.1506     | **0.3300** | **0.2934** | **0.6403** |
> | `GPT‑4o`      |   |   |   |   |   |
> | State  | 7.8105     | 0.1711     | 0.5178     | 0.2388     | 0.5962     |
> | Recent | 7.2690     | 0.1607     | 0.4576     | 0.2574     | 0.6200     |
> | Popular| 8.8492     | 0.1818     | 0.5198     | 0.2138     | 0.5798     |
> | **MF‑LLM** | **2.2669** | **0.1085** | **0.2211** | **0.4103** | **0.7249** |
> | `DeepSeek‑R1‑Distill‑Qwen‑32B` |   |   |   |   |   |
> | State  | 1.7496     | 0.1105     | 0.2847     | 0.4173     | 0.6999     |
> | Recent | 2.4299     | 0.1192     | 0.3462     | 0.3941     | 0.6829     |
> | Popular| 1.4311     | **0.0905** | 0.2198     | 0.4424     | **0.7256** |
> | **MF‑LLM** | **1.1814** | 0.0922     | **0.2042** | **0.4483** | 0.7204     |
>
> ------
> > **W2: "Scalability is cited as a main motivation, yet the paper lacks any complexity analysis or detailed experiments on efficiency and scalability."**
>
> **(1) Detailed experiments on Scalability**
>
> We also **extended the simulation length** (Qwen2‑1.5B‑Instruct, without fine‑tuning) to the equivalent of **4000** agent updates (previously reported as “steps”). **Results** for 500–4000 agents are shown below, confirming stable performance over long horizons.
>
> | Agents | KL↓    | WD↓    | DTW↓   | NLL    | Macro F1↑ | Micro F1↑ |
> | - | - | - | - | - | - | - |
> | 500   | 1.2582 | 0.0645 | 0.1196 | 3.9916 | 0.3688    | 0.8644    |
> | 1000  | 1.4638 | 0.0729 | 0.1220 | 4.0112 | 0.3643    | 0.8436    |
> | 2000  | 1.1053 | 0.0685 | 0.0965 | 4.0236 | 0.3569    | 0.8537    |
> | 3000  | 1.0668 | 0.0650 | 0.0837 | 4.0142 | 0.3592    | 0.8611    |
> | 4000  | 1.5905 | 0.0698 | 0.1025 | 4.0237 | 0.3548    | 0.8376    |
>
> **(2) Detailed experiments on complexity analysis**
>
> To complement the theoretical **O(N) complexity analysis** presented earlier, we **conducted empirical measurements** to verify runtime scaling in practice. We measured simulation runtime for varying numbers of agent updates on `Qwen2‑1.5B‑Instruct` (Twitter dataset, without fine‑tuning). Experiments were run on a single NVIDIA A100 GPU (40 GB memory).
>
> | Agents | Time (s) | Time / Agent (s) |
> |- | - | - |
> | 512    | 280.88   | 0.55 |
> | 1008   | 561.89   | 0.56   |
> | 1504   | 832.31   | 0.55|
> | 2000   | 1113.41  | 0.56|
> | 2512   | 1404.34  | 0.56|
> | 3008   | 1687.50  | 0.56|
> | 3504   | 1966.29  | 0.56|
> | 4000   |2249.15  | 0.56|
>
> **Finding:** Runtime increases **linearly** with the number of agents, maintaining a nearly constant time per agent (~0.56 s), which primarily reflects the cost of repeated LLM calls. This empirical result validates the theoretical **O(N)** complexity, providing experimental evidence for MF‑LLM’s scalability.
>
> ------
>
> **We hope that these efforts will alleviate your concerns regarding MF‑LLM. Your feedback is highly valuable to us, and we would appreciate any updates or further guidance you might have regarding our revisions and responses.**

---

> > ### Comment · Reviewer_G3Ud · 2025-08-05
> >
> > The rebuttal is convincing, and I have adjusted my score.

---

> > > ### Author Response · Authors · 2025-08-05
> > > **Thank You for Recognizing Our Convincing Rebuttal**
> > >
> > > **Dear Reviewer G3Ud,**
> > >
> > > We sincerely appreciate your recognition of our convincing rebuttal and your adjusted score. We are encouraged by your support and will integrate all additional results and analyses into the revised paper to ensure the final version is as clear and rigorous as possible.
> > >
> > > Thank you again for your thoughtful engagement, and we would be grateful for your continued support during the upcoming AC–reviewer discussion period.

---

### Official Review · Reviewer_U3Se · 2025-07-02

**Clarity:** 2
**Significance:** 2
**Originality:** 3
**Rating:** 5
**Confidence:** 3

**Summary:**

This is a work on simulation of agent-based models of human social interaction using large language model agents.  The focus is on modeling social networks and phenomena such as the spread of rumors and collective decision making. The present work builds on the framework of Refs. [32] and [43], it studies the same task of using a stylized social network which generates a time sequence of "tweets" to model the Weibo corpus (Ref [24]), and it compares with previous results.  Two innovations are claimed.  The first is to use an LLM (the "mean field" agent) to summarize the vector of states and actions for all of the agents to produce a "mean field" summary text.  My understanding is that direct interactions between the agents (the "tweets") are replaced by the summary text.  The second innovation is to introduce a loss based on information bottleneck to train the mean field agent, balancing effectiveness of the summary in making subsequent predictions against compression of the information being summarized.  The results are compared with several ablations (leave out specialized training, leave out the summary) and the new method achieves a better fit.

**Questions:**

1. Given that the mean field summary depends on the states of all of the agents (equation 1), and considering that the actual social systems being modeled as well as the baselines do not (as I understand) include direct communication of the agents' states, how do we know whether the reported improvements in predictive fit might be because of this new and unrealistic form of interaction ?

2. The paper includes examples of agent comments, but I didn't see any examples of mean field summaries.  This would help answer (1).

2. Is there any estimate of the uncertainty and statistical significance of the results in tables 1,2 ?

**Ethical Concerns:**

["NO or VERY MINOR ethics concerns only"]

**Final Justification:**

The authors responded to my comments and questions adequately.

**Limitations:**

I think it is adequate.  The study is methodological in nature so possible negative impact would be speculative.

**Quality:**

3

**Strengths And Weaknesses:**

Strengths: adding the summary is an interesting idea which is probably new in this context and certainly has not been much explored.

Weaknesses: I found the description of the simulation difficult to read and understand.  Basic attributes such as the connectivity of the network (are all agents connected pairwise or not?) are not clearly explained.  Especially, it was not clear whether the summary is used to *replace* agent-agent interactions (as stated on p. 2) or *in addition* to the other interactions listed under "baselines" on p. 6.   I guess that it is used to replace, but this seems like a very drastic change in the model compared to the previous work.  One also wonders whether the improvements in prediction are coming because the summary contains information about the agents' internal states which is not being communicated either in the real world social interactions being modeled or in the baselines.  In this case it would be a sort of contamination or "cheating" which would make the results less interesting.

---

> ### Author Rebuttal · Authors · 2025-07-31
>
> **Dear Reviewer U3Se,**
>
> Thank you very much for taking the time to review MF‑LLM and for your constructive feedback, as well as for recognizing the “interesting idea” and “better fit.”
>
> During the rebuttal period, we have made every effort to address your concerns through targeted clarifications, additional experiments, and concrete examples:
>
> - **Clarified the simulation mechanism and connectivity**.
> - **Added new case studies** on *Different Interactions* and *Different Users*.
> - **Provided explicit mean field summaries**, showing agent actions and corresponding updated mean field content.
> - **Demonstrated that the states in Eq. (1) are observable from real‑world data**, not unobservable “internal states.”
> - **Conducted control experiments by adding state information to baselines**, with MF‑LLM still outperforming all variants.
> - **Reported statistical significance** with mean ± standard deviation over multiple runs.
>
> Below, we provide detailed responses to each of your questions, with the utmost sincerity, to fully address every concern.
>
> ---
>
>
>
> > **W1 & Q2: Is the mean field summary used to replace agent-agent interactions or in addition to the interactions listed under 'baselines'? Could you provide examples of mean field summaries?**
>
> **Answer:** We thank the reviewer for the insightful comments. We will clarify the simulation description in the revised paper.
>
> **1. Mechanism:**
>
> Many existing works model the connectivity of the agent network, where each edge weight reflects the strength of influence between two agents. **While such network modeling is reasonable, manually specifying these edge weights is often inaccurate and may overlook important interaction patterns in unseen scenarios.**
>
> MF‑LLM replaces fixed, handcrafted connectivity with a **data-driven mechanism**:
>
> - The **policy model** generates individual actions based on the agent’s state and the current population signal.
> - The **mean field model** updates the population signal using these newly generated actions
>
> This creates an indirect but generalizable interaction pathway:
> $$a_A→m_{t+1}→a_B$$
> where agent A’s action affects agent B via the updated population signal. Crucially, interaction strengths are **inferred and updated dynamically at each step from online data**, without any manually set network weights.
>
> The baselines represent alternative agent interaction strategies:
>
> - **Recent** connects agents to the most recent actions.
> - **Popular** connects agents to the most popular actions.
>
> These fixed heuristics may miss subtle interactions in new contexts. In contrast, MF-LLM allows interaction patterns to emerge adaptively from the data.
>
>
>  **2. Experimental Evidence:**
>
> Our MF-LLM **do not** force the mean field model to attend to or ignore specific agents. It evaluates all available information to infer interaction effects. **The following two case studies illustrate this dynamic modeling, while also showcasing the content of the mean field**.
>
> **(1) Influence of Different Interactions**
>
> In a *rumor propagation* scenario, we compare the effects of two different actions taken by the same **high-influence user**.
>
> - **State**: A user with high influence and many friends.
>
> - **Previous Mean Field**:
>
>   `Most users express opposition and criticism toward the event, regarding it as unfair competition.`
>
> - **Action A**: `"Repost"`
>
> - **Action B**: `"Looking forward to the truth!"`
>
>
>
> The mean field model takes the individual’s state and action as input and **updates the mean field** accordingly:
>
> **Updated Mean field:**
>
> - **After Action A**:
>
>   `"Most users are criticizing the event rather than seeking factual verification."`
>
> - **After Action B**:
>
>   `"Users call for an investigation to uncover the cause. … Many suspect that this might be a genuine news report."`
>
> Due to rebuttal length limits, only excerpts of updated mean field are shown here; the full version will appear in the revised paper.
>
>
>
> **(2) Influence of Different Users**
>
> In the *rumor propagation* scenario, we compare the effects of the **same action** taken by two users: one with **high** influence and one with **low** influence.
>
>
>
> - **Action**: `"Looking forward to the truth!"`
> - **Previous Mean Field**: *(Same as in Case Study 1)*
>
>
>
> - **Agent States**:
>
>   - **High-Influence User**: A user with high influence and many friends.
>   - **Low-Influence User**: Same description, but with low influence and few friends.
>
> **Updated Mean Field:**
>
> - After **High-Influence User** acts: `"Many suspect that this might be a genuine news report."`
>
> - After **Low-Influence User** acts: `"A few users in the comments express a desire to uncover the truth."`
>
>
>
> **Conclusion: MF‑LLM thus provides an indirect yet generalizable mechanism for modeling agent interactions and influence, without manually specifying network weights.**
>
> ---
>
>
>
> > **W2 & Q1: "How do we know whether the reported improvements in predictive fit might be because of this new and unrealistic form of interaction?"**
>
> **Answer:** We thank the reviewer for raising this concern.
>
> 1. **Observed states in Equation 1.** The “state” in Eq. (1) corresponds to **information observable in the real-world social interactions** being modeled. These states include attributes such as `location, description, gender, friend/follower counts, activity level, and verification status`. The *User State Construction* prompt template is shown in **Appendix D.4 (p. 20)** of the submitted paper. No hidden or unobservable “internal states” are introduced.
>
>
> 2. **The mean field model processes these observable states, which is fully reasonable**. As illustrated in above Case Study 2 (*Influence of Different Users*), even when different users perform the same action, their states lead to different downstream effects on subsequent users. Processing these observable states enables the mean field model to analyze the corresponding influence of actions more accurately.
>
>
> 3. **We conducted a control experiment by adding user state information to the baselines (Recent, Popular).** Recent (+state) improves, while Popular (+state) drops in performance. **MF-LLM still outperforms these enhanced baselines,** confirming that the improvement comes from its interaction modeling. Baselines ignoring user identity fail to capture heterogeneous agent influence, highlighting MF-LLM’s advantage.
>
> | **Model**  | **KL↓**   | **WD↓** | **DTW↓** | **Macro F1↑** | **Micro F1↑** |
> | - | - | - | - | - | - |
> | Recent (original)| 1.277±0.670| 0.077±0.008 | 0.202±0.060 | 0.441±0.032   | 0.799±0.026   |
> | Recent (+ state) | **0.9038±0.495** | **0.0664±0.012** | **0.1674±0.064** | **0.4667±0.024** | **0.8273±0.030** |
> | Popular (original)| 1.288±0.579 | 0.080±0.007| 0.199±0.055 | 0.435±0.032   | 0.792±0.025   |
> | Popular (+ state)| 1.4672±0.687   | 0.0821±0.017 | 0.2087±0.073   | 0.4358±0.023  | 0.7861±0.037  |
> | MF‑LLM  | 0.645±0.260    | 0.065±0.007 | 0.152±0.020 | 0.486±0.023   | 0.839±0.009 |
> | MF‑LLM IB‑Tune  | **0.512±0.238**| **0.062±0.007** | **0.138±0.025**| **0.495±0.024**| **0.846±0.009**|
>
>
> ---
>
> > **Q3: "Is there any estimate of the uncertainty and statistical significance of the results in tables 1,2?"**
>
> **Answer:** We report **mean ± standard deviation and p-values** for Tables 1 and 2. The small standard deviations and consistent performance of MF‑LLM indicate statistical significance and stability.
>
> Due to rebuttal length limits, we only present the main results here; the full version will be included in the revised version.
>
> **Tables 1 - mean ± std:**
>
> | **Model** | **KL↓**   | **WD↓** | **DTW↓** |
> | - | - | - | - |
> | GPT-4o-mini | | | | | |
> | State | 4.172±2.618 | 0.127±0.022 | 0.420±0.092 |
> | Recent | 6.728±1.525 | 0.145±0.016 | 0.485±0.072 |
> | Popular | 5.591±2.298 | 0.128±0.017 | 0.447±0.096 |
> | MF-LLM | **1.647±0.702** | **0.100±0.017** | **0.329±0.118** |
> | DeepSeek-R1-32B ||||
> | State | 1.946±0.981 | 0.110±0.016 | 0.349±0.074 |
> | Recent | 6.435±1.226 | 0.141±0.018 | 0.495±0.105 |
> | Popular | 3.812±1.427 | 0.113±0.011 | 0.385±0.110 |
> | MF-LLM | **1.280±0.831** | **0.088±0.015** | **0.307±0.101** |
> | Qwen2-7B-Instruct | | | |
> | State | 1.153±0.667 | 0.085±0.012 | 0.239±0.077 |
> | Recent | 1.346±0.184 | 0.081±0.010 | 0.218±0.059 |
> | Popular | 1.066±0.378 | 0.076±0.012 | 0.199±0.062 |
> | MF-LLM | **1.010±0.354** | **0.075±0.014** | **0.198±0.068** |
> | Qwen2-1.5B-Instruct | | | |
> | State | 0.966±0.491 | 0.068±0.009 | 0.166±0.046 |
> | Recent | 1.277±0.670 | 0.077±0.008 | 0.202±0.060 |
> | Popular | 1.288±0.579 | 0.080±0.007 | 0.199±0.055 |
> | MF-LLM | 0.645±0.260 | 0.065±0.007 | 0.152±0.020 |
> | MF-LLM IB-Tune | **0.512±0.238** | **0.062±0.007** | **0.138±0.025** |
>
> Due to character limits, we present the KL Divergence statistical significance for the Qwen2-1.5B-Instruct model. The following p-values highlight **significant differences** between algorithms:
>
> | **Algorithm Pair**| **p-value**|
> | -- | - |
> | Popular vs MF-LLM| **0.0095** (*) |
> | Popular vs Recent| 0.9718|
> | Popular vs State | 0.2236|
> | Popular vs MF-LLM IB-Tune | **0.0008** (*) |
> | MF-LLM vs Recent | **0.0132** (*) |
> | MF-LLM vs State  | 0.1293|
> | MF-LLM vs MF-LLM IB-Tune  | 0.4063|
> | Recent vs State  | 0.2510|
> | Recent vs MF-LLM IB-Tune| **0.0014** (*) |
> | State vs MF-LLM IB-Tune | **0.0179** (*) |
>
> (*) p-values < 0.05 indicate statistically significant differences.
>
> **Tables 2:**
>
> | **Method** | **KL↓** | **WD↓** | **DTW↓** |
> | - | - | - | - |
> | MF-LLM (Ours) | **0.481±0.236** | **0.070±0.018** | **0.158±0.042** |
> | w/o IB-Tune MF|0.531±0.212 | 0.070±0.018 | 0.159±0.045|
> | w/o IB-Tune Policy| 0.588±0.239 | 0.070±0.018 | 0.158±0.048 |
> | w/o IB-Tune|0.617±0.250 | 0.072±0.021 | 0.164±0.045 |
> | SFT (no MF)| 0.753±0.556 | 0.075±0.022 | 0.186±0.074 |
> | Pretrained (no MF) | 1.053±0.760 | 0.079±0.020 | 0.195±0.068 |
>
> ---
>
> **We hope these clarifications and new results will encourage you to reconsider your score. If you have any further questions, we are happy to discuss them.**

---

### Official Review · Reviewer_GZm3 · 2025-07-02

**Clarity:** 4
**Significance:** 3
**Originality:** 3
**Rating:** 5
**Confidence:** 3

**Summary:**

This paper introduces the Mean-Field LLM (MF-LLM), a novel framework for simulating population-level decision dynamics. The core idea is to integrate mean-field theory into a large language model (LLM)-based agent simulation. This approach models the bidirectional feedback loop where individual decisions are influenced by an aggregate population signal (the "mean field"), and these decisions, in turn, update that signal. This replaces computationally expensive agent-to-agent interactions with a more scalable agent-to-population model. To improve the simulation's alignment with real-world data, the authors propose IB-Tune, a fine-tuning algorithm inspired by the Information Bottleneck principle. IB-Tune optimizes the LLM modules to generate population signals that are predictive of future actions while filtering out redundant historical information. The framework is evaluated on a large real-world social media dataset, demonstrating significant improvements in quantitative accuracy and forecasting capabilities compared to existing methods.

**Questions:**

The mean-field approximation is key to scalability. Could you elaborate on the types of social phenomena or network structures where this approximation might break down?

The finding that smaller LLMs outperform larger ones due to response homogeneity is interesting. Have you experimented with decoding parameters for the larger models (e.g., increasing temperature, top-p sampling) to see if encouraging more diversity could improve their performance in your framework?

The IB-Tune objective in Equation 7 includes a hyperparameter β, which balances compression and prediction. How sensitive is the model's performance to this parameter? Could you provide some details on how it was chosen and its impact on the learning process?

The framework uses a "warm-up" phase with real data before the simulation phase begins. How does the duration of this warm-up affect the long-term fidelity of the simulation? Is there a noticeable "drift" from the real-world trajectory, and does it correlate with the length of the warm-up?

**Ethical Concerns:**

["NO or VERY MINOR ethics concerns only"]

**Final Justification:**

This paper combines mean-field theory with LLM-based simulation to create a scalable alternative to agent-to-agent interaction. The method is technically strong, and the IB-Tune algorithm offers a clear and effective way to align simulations with real-world data. The experiments use a large, real-world dataset and include strong baselines, many evaluation metrics, and clear ablation studies showing the importance of each part of the method. The paper is well written, reproducible, and openly discusses its limitations. Although automatic detection of external events and deeper study of the performance difference between small and large LLMs are still missing, the overall strengths are much greater than these issues. I recommend acceptance.

**Limitations:**

The authors have clearly acknowledged and discussed the limitations of their work. In Section 5 and Appendix C, they identify important challenges. One issue is the accumulation of small errors over long simulation periods, which can cause the overall behavior of the system to drift away from realistic patterns. They also recognize that the current framework requires manual input of external signals, rather than detecting these exogenous events automatically. This limits the framework's usefulness in real-time or fully automated forecasting settings. In addition, the authors offer a thoughtful explanation for the surprising result that smaller LLMs sometimes perform better than larger ones. They connect this to the idea that smaller models may produce more diverse behaviors, which is important for simulating complex social systems. In general, the discussion is honest and clear, and it provides useful directions for future work.

**Paper Formatting Concerns:**

None.

**Quality:**

3

**Strengths And Weaknesses:**

Strengths
This paper presents a new and important idea by combining mean-field theory with large language model (LLM) based social simulation. This is a big improvement over older methods, which often use simple prompts or fine-tuning methods that do not consider how agents interact with each other. The proposed method, called MF-LLM, focuses on the feedback loop between individual agents and the whole population. This helps to understand how group behaviors appear and change over time. Solving this problem poses a challenge in computational social science, and MF-LLM provides a clear and scalable solution. The method could be useful in real-world applications such as predicting public opinion and planning social interventions.
The technical design is strong and well-structured. MF-LLM includes two LLM modules that work in turns: a policy model and a mean-field model. This structure is easy to understand and helps to capture how individuals influence the group and how the group influences individuals. Another key part is the IB-Tune algorithm, which uses the Information Bottleneck principle. It fine-tunes the models in a data-driven way, focusing on keeping only the most important parts of the population state that help predict future actions. This is a more advanced and effective method for aligning data compared to standard fine-tuning.
The paper is very clear and well organized. Figure 1 gives a helpful overview of how the full system works. The authors clearly explain the three main challenges they focus on: aligning data, modeling interaction dynamics, and scaling the method. They also show how their method addresses each of these problems. The appendices provide extra details such as prompt templates, hyperparameters, and additional results, which help readers better understand the work and reproduce the experiments.
The experiments are complete and convincing. The authors use a large and real-world dataset called WEIBO, which is better than using only simple or artificial data. They compare their method with strong baseline models and utilize a wide range of evaluation metrics, including KL Divergence, Wasserstein distance, Dynamic Time Warping, and F1 scores. These metrics help measure how well the model works in both time and meaning. The results are interesting. The model reduces KL Divergence by 47% compared to versions that do not use the mean-field module, and it performs well across seven different domains and four LLM backbones. The ablation study in Table 2 also shows clearly that both the mean-field module and the IB-Tune algorithm are very important. When the mean-field module is removed, performance drops by as much as 118%

Weaknesses
First, the mean-field approximation, although necessary to make the model scalable, reduces all interactions to a simple relationship between each agent and the overall population. This approach may miss more complex and local interactions, such as the spread of influence within small, close-knit groups or the impact of specific individuals who are highly influential. These types of local behaviors are important in many real-world social systems and may not be fully captured by the current model.

Second, the authors show that adding external signals, called exogenous events, can help improve the accuracy of the simulation. However, this process is still done manually. For real-world forecasting tasks, this is a major limitation. In many real situations, external events such as news, policy changes, or viral content happen suddenly, and the system needs to detect and include them automatically. Without this ability, the model may miss important changes in social dynamics.

Lastly, the paper presents an interesting and unexpected result: smaller LLMs sometimes perform better than larger ones. The authors suggest that this may be because larger models tend to produce more similar and less diverse responses. While this explanation makes sense, it would be more convincing if the authors included a deeper analysis. For example, testing different decoding methods such as increasing the temperature in larger models could show whether the lack of diversity is due to model size or decoding choices. More evidence would help support this claim and provide useful guidance for model selection.

---

> ### Author Rebuttal · Authors · 2025-07-31
>
> **Dear Reviewer GZm3**,
>
> Thank you for your detailed and constructive review, and for recognizing our work with a *“strong and well‑structured” technical design, “complete and convincing” experiments, and a “clear and well‑organized” paper.*
>
> During the rebuttal, we have made every effort to address your comments through additional experiments, clarifications, and expanded analysis:
>
> - **Added case studies** of highly influential individuals (Appendix C, Fig. 8‑4) and small, close‑knit groups, demonstrating that MF‑LLM effectively captures complex interactions without manual design.
> - **Added new decoding experiments** (varying temperature and top‑p) and highlighted Appendix G outputs showing lexical homogeneity in larger LLMs.
> - **Conducted sensitivity analysis** over multiple β values, showing the relationship between β and loss values.
> - **Demonstrated that** longer warm‑up phases improve long‑term simulation fidelity (Appendix C, Fig. 5a).
>
> **Below, we provide detailed responses to each of your questions, with the utmost sincerity, to address every concern.**
>
>
>
> ---
>
>
>
> > **W1 & Q1: "Mean-Field Approximation ... may miss more complex and local interactions"?**
>
> **Answer:** We respectfully clarify that MF‑LLM **does not** “miss more complex and local interactions” or “break down” in such cases. MF‑LLM is evaluated on real‑world data, which naturally includes "small, close‑knit groups" and "highly influential individuals". We address this via mechanism and empirical evidence:
>
> **1. Mechanism**
>
> In MF‑LLM, the **policy model** maps population signals to individual actions, while the **mean field model** updates the population signal from new actions. The pathway
> $$a_A→m_{t+1}→a_B$$
> represents an indirect but generalizable interaction between any pair of agents. Unlike prior work with fixed influence weights, MF‑LLM **infers and updates interaction effects dynamically from online data**, enabling robust modeling even in unseen structures.
>
> **2. Empirical Evidence**
>
> **(a) Highly influential individuals**
> **Appendix C and Fig. 8(4) show** "impact of highly influential individuals": when the famous Chinese sports commentator "@Huang Jianxiang" was referenced to verify the truth, the population signal shifted toward skepticism, leading to reduced rumor spread.
>
> **(b) Small, close‑knit groups**
> In an entertainment discussion:
>
> - **User A**: `"Taylor Swift will hold a concert!" `
> - **User B**: `"The city marathon route will be adjusted next month."`
>
> **Updated Mean Field (after A+B):**
> `"Population sentiment surges around the concert... marathon news draws moderate interest from runners."`
>
> **Policy model responses under same mean field:**
>
> - **Idol Fan User**: `"Can’t believe it! Already organizing a group to buy tickets and cheer!"  `
> - **Marathon Enthusiast**: `"Good to know about the route change—I’ll adjust my training."`
>
> This shows MF‑LLM captures topic‑specific shifts via the mean field and generates heterogeneous actions via the policy model—faithfully modeling "close‑knit group" dynamics from real data.
>
>
>
>
>
> ---
>
> > **W2: Although the authors show that adding external signals improves simulation accuracy, how does the model handle the automatic detection in real-world scenarios?**
>
> **Answer:** We appreciate the reviewer’s recognition of our “honest and clear” limitation discussion. **Appendix C & Fig. 8 shows that** adding external signals preserves simulation fidelity, demonstrating MF‑LLM’s robustness. While automatic detection is beyond our current simulation-focused scope, we agree it is important for real-world forecasting and identify it as a future direction.
>
>
>
>
> ---
>
> > **W3 & Q2: Can you provide more evidence for why smaller LLMs outperform larger ones, such as by exploring decoding parameters?**
>
> **Answer:** Regarding this "interesting and unexpected result", **we discussed it in Appendix G and presented a comparison of model outputs** (pp. 27–28 for larger LLMs vs. pp. 25–26 for smaller LLMs).  The larger LLMs’ generations exhibit clear lexical homogeneity—frequent repetition of similar phrasing and syntactic structures (highlighted by color)—whereas the smaller LLMs, under the same simulation context, show no such degree of lexical uniformity.
>
> Further, **we conducted additional experiments varying decoding parameters for the larger models**. The following table presents the results for:
>
> 1. **Higher temperature** (temp = 1.2, top‑p = 0.85)
> 2. **Higher top‑p** (temp = 0.8, top‑p = 0.95)
>
> For each metric, we report the absolute value together with its relative change (%) compared to the original decoding setup in the submitted paper (temperature = 0.8, top‑p = 0.85).
>
> *7B = Qwen2‑7B‑Instruct*
>
> | Model       |    Temp |    Top_p | KL↓               | WD↓              | DTW↓             | MacroF1↑         | MicroF1↑         | NLL↓             |
> | :---------- | ------: | -------: | :---------------- | :--------------- | :--------------- | :--------------- | :--------------- | :--------------- |
> | 7B          |     0.8 |     0.85 | 0.915             | 0.098            | 0.332            | 0.388            | 0.705            | 3.994            |
> | 7B          | **1.2** |     0.85 | **0.814(-11.0%)** | **0.098(-0.6%)** | **0.328(-1.2%)** | **0.389(+0.4%)** | **0.715(+1.5%)** | **4.044(+1.3%)** |
> | 7B          |     0.8 | **0.95** | 1.960(+114.3%)    | 0.116(+18.3%)    | 0.413(+24.6%)    | 0.362(-6.7%)     | 0.670(-5.0%)     | 4.076(+2.0%)     |
> | GPT‑4o‑mini |     0.8 |     0.85 | 0.990             | 0.106            | 0.357            | 0.380            | 0.687            | -                |
> | GPT‑4o‑mini | **1.2** |     0.85 | 2.673(+169.9%)    | 0.128(+20.6%)    | 0.467(+30.9%)    | 0.337(-11.3%)    | 0.636(-7.4%)     | -                |
> | GPT‑4o‑mini |     0.8 | **0.95** | 2.724(+175.1%)    | 0.132(+23.6%)    | 0.477(+33.7%)    | 0.337(-11.3%)    | 0.633(-7.9%)     | -                |
>
> These results show that increasing diversity through **higher temperature or top‑p does not necessarily improve simulation performance**. For example, Qwen2‑7B‑Instruct showed only a marginal improvement when increasing temperature, but significant degradation at top‑p = 0.95 (KL more than doubled). Similarly, GPT‑4o‑mini exhibited worse performance when either temperature or top‑p was increased.
>
> Due to rebuttal time constraints, we report the additional results above. More extensive experiments on “increasing diversity” will be included in the revised paper for reference.
>
> ---
>
>
>
> > **Q3: How sensitive is the model's performance to the β in the IB-Tune objective?**
>
> **Answer:** We thank the reviewer for raising the question regarding the sensitivity of hyperparameter β in balancing compression and prediction.
>
> In our experiments, **we tested β values of 1.2, 1.6, and 2.0** and observed that the predictive term **$- \sum_{i=1}^{N_t} \log \pi(a_t^{i} \mid s_t^i, m_t)$** (denoted as $L_2$ hereafter) in Equation 7 shows a qualitative negative correlation with β. The following table summarizes the average $L_2$ over the final 100 steps of training:
>
> | **β** | **$L_2$** |
> | ----- | --------- |
> | 2.0   | 3.563     |
> | 1.6   | 3.978     |
> | 1.2   | 3.987     |
>
> As shown, a lower $L_2$ indicates better predictive performance. Based on these qualitative observations, we selected **β = 2** for the final model.
>
> We have already begun testing larger β values in additional experiments. However, due to the extensive fine-tuning time required for LLMs, we aim to obtain results before the discussion period ends. We will include these findings in the revised version of the paper.
>
>
>
> ---
>
>
>
> > **Q4: How does the warm-up duration affect the long-term fidelity of the simulation?**
>
> **Answer:** We thank the reviewer for the insightful question regarding the warm‑up phase.
>
> As discussed in **Appendix C** and shown in **Fig. 5(a)**, we evaluated simulations starting at different warm‑up lengths (20, 34, 43, 70 steps). **Longer warm‑up phases consistently yielded smaller deviation from the real‑world trajectory.** The figure also marked the prediction and real values, directly illustrating this improvement. This benefit arose because a longer warm‑up provided the simulation with a richer and more accurate prior, reducing early‑stage uncertainty.
>
>
> ---
>
> **We hope these clarifications and new results will address your concerns. If you have any further questions, we are happy to discuss them.**

---

> > ### Comment · Reviewer_GZm3 · 2025-08-01
> >
> > Dear authors,
> >
> > I truly appreciate the effort you put into answering all my questions and comments. I firmly believe that your work is important to the public health and community's research. In this field, every strong contribution counts, especially when researchers apply novel ideas to understand population behaviors and they intend to improve their quality of life. I will keep the rating of 5.

---

> > > ### Author Response · Authors · 2025-08-02
> > > **Sincere thanks for your recognition and support of our work.**
> > >
> > > **Dear Reviewer**,
> > >
> > > Thank you sincerely for your support and encouragement. Your feedback has greatly motivated us to further refine and extend this research. We will continue striving to deliver meaningful insights into population behaviors and their implications for improving quality of life.

---

### Official Review · Reviewer_f92d · 2025-07-03

**Clarity:** 3
**Significance:** 2
**Originality:** 3
**Rating:** 5
**Confidence:** 4

**Summary:**

This paper introduces a framework to integrate mean field theory with LLMs for simulating population decision-making dynamics,. The authors develop IB-Tune, a  fine-tuning algorithm inspired by Information Bottleneck principles that optimizes both a mean field model and a policy model to generate realistic individual decisions. Evaluated on ~4,500 real-world events from the WEIBO corpus, MF-LLM reduces KL divergence by 47% compared to baselines and accurately captures population trends across multiple domains and LLM backbones, enabling applications like trend forecasting and intervention planning.

**Questions:**

- How does the mean field approximation handle scenarios where agent interactions are highly heterogeneous or when certain agents have disproportionate influence on population dynamics?
- How would you validate MF-LLM's performance on entirely different cultural contexts where social interaction norms differ significantly?
- How do you address potential bias from using GPT-4o-mini for semantic evaluation? Have you validated these automated assessments against human expert judgments?

**Ethical Concerns:**

["NO or VERY MINOR ethics concerns only"]

**Final Justification:**

The authors' rebuttal with extensive additional experiments has resolved my primary concerns about cultural generalizability (demonstrated through English Twitter datasets and market behavior simulation), scalability (empirical O(N) scaling up to 4000 agents), and evaluation methodology (91.25% human expert agreement validation). The controlled experiments on heterogeneous interactions and disproportionate influence effectively address the mean field approximation concerns, while cross-cultural validation demonstrates robust performance beyond the original WEIBO corpus.

**Limitations:**

The paper's scalability claims are undermined by the absence of computational complexity analysis and testing on truly large populations, with simulations extending only 900 timesteps and acknowledged compounding errors over time. While positioned as achieving "quantitative alignment with real-world data," the framework is validated exclusively on WEIBO corpus discussions, raising questions about generalizability to other forms of collective decision-making like voting or market behavior that motivated the work. Despite testing across seven domains and four LLM backbones, all evaluation occurs within a single cultural context, and the mean field approximation may oversimplify real social dynamics by failing to capture heterogeneous interaction patterns and varying influence weights between agent types that are crucial in authentic social systems.

**Paper Formatting Concerns:**

No issues

**Quality:**

2

**Strengths And Weaknesses:**

Strengths
- The integration of mean field theory with LLMs is  theoretically grounded, providing a principled approach to modeling population dynamics while maintaining computational scalability.
The paper demonstrates robust performance across multiple dimensions - 7 domains, 4 LLM backbones, 8 semantic dimensions, and ~4,500 real-world events, with consistent 47% improvement in KL divergence over baselines.
Weakness
- Evaluation is only restricted to the WEIBO corpus and thus very specific to the context within that corpus.
Although

---

> ### Author Rebuttal · Authors · 2025-07-31
>
> Dear Reviewer f92d,
>
> Thank you for your detailed review and valuable feedback, and for recognizing MF‑LLM as a *“theoretically grounded”* approach with *“robust performance.”*
>
> During the rebuttal, we conducted extensive experiments and analyses to address your comments:
>
> - **Added experiments** on culturally diverse English datasets (Twitter) and market behavior simulation, confirming MF-LLM’s generalization cross different cultural contexts.
> - **Provided mechanism explanation, empirical results** (Appendix C, Fig.8) and **new case studies**, showing MF-LLM handles heterogeneous interactions and disproportionate influence.
> - **Added comparison** of GPT‑4o‑mini with human experts, GPT-4, GPT-4o, GPT-3.5, confirming it is the best balance of accuracy, cost, and speed for semantic evaluation.
> - **Added computational complexity analysis** (O(N²)→O(N)) and **long-horizon experiments** (Twitter, 2000 steps), demonstrating scalability.
>
> **Here are our detailed responses to each of your questions and suggestions. We have made every effort, with the utmost sincerity, to address every one of your concerns.**
>
> ---
>
>
>
> > **Weakness & Q2: "How would you validate MF-LLM's performance on entirely different cultural contexts where social interaction norms differ significantly?"**
>
> **Answer:** Thank you for this important question. MF-LLM is inherently culture-agnostic, with no assumptions about language or cultural background. To validate its generalization ability, we added experiments on **culturally diverse English datasets** without domain-specific fine-tuning.
>
> **1. Famous Keyword Twitter Replies Dataset** (Kaggle)
>
>  This dataset covers globally relevant topics (COVID, Vaccine, NFT, TikTok, WorldCup, etc.). Using Qwen2-1.5B-Instruct, MF-LLM consistently outperforms baselines:
>
> | Method     | KL↓        | WD↓        | DTW↓       | NLL↓       | Macro F1↑  | Micro F1↑  |
> | - | - | - | - | - | - | - |
> | State      | 0.8696     | **0.0673** | 0.1700     | 4.2121     | 0.4977     | **0.8241** |
> | Recent     | 1.5061     | 0.0922     | 0.2206     | 4.1053     | 0.4471     | 0.7661     |
> | Popular    | 1.0059     | 0.0725     | 0.1553     | 4.0697     | 0.4864     | 0.8047     |
> | **MF-LLM** | **0.6883** | 0.0744     | **0.1587** | **3.9748** | **0.4989** | 0.8093     |
>
> **2. Market Behavior Simulation**
>
>  To assess MF-LLM’s generalization to market behavior, we refer to *TwinMarket* [1], which uses real-world transaction data from Chinese social platforms (Xueqiu, Guba), and *ElectionSim* [2], which uses Twitter data on elections. However, neither releases datasets, making direct comparison infeasible.
>
> Following their approach of using social media data to simulate collective decisions, we use Bitcoin-related tweet–reply data from Twitter to simulate dynamic **bullish/bearish sentiment** at the population level. Once again, MF-LLM achieves superior alignment:
>
> | Method     | KL↓        | WD↓        | DTW↓       | NLL↓       | Macro F1↑  | Micro F1↑  |
> | - | - | - | - | - | - | - |
> | State      | 1.2304     | 0.0854     | 0.4505     | 4.4320     | 0.4723     | 0.6721     |
> | Recent     | 2.0641     | **0.0800** | 0.4730     | 4.3438     | 0.4550     | 0.6573     |
> | Popular    | 1.4394     | 0.0853     | 0.3930     | 4.2537     | 0.4834     | 0.6920     |
> | **MF-LLM** | **0.9316** | 0.0915     | **0.2364** | **4.1485** | **0.5025** | **0.7741** |
>
> We hope these additional results address your concern by showing that MF-LLM generalizes across languages and cultural settings. These findings will be incorporated into the revised paper.
>
> [1] TwinMarket: A Scalable Behavioral and Social Simulation for Financial Markets.
>
> [2] ElectionSim: Massive Population Election Simulation Powered by LLM Agents.
>
>
>
>
>
> ---
>
>
>
> > **Q1: "How does the mean field approximation handle scenarios where agent interactions are highly heterogeneous or when certain agents have disproportionate influence on population dynamics?"**
>
> **Answer:** First, we clarify that **MF-LLM is evaluated on real-world data**, which naturally involves abundant heterogeneous agent interactions and cases of disproportionate influence. Below, we address this from both a **mechanism** and **empirical evidence** perspective:
>
> **1. Mechanism**: MF-LLM places no assumptions on agent homogeneity. At each step, the policy model generates personalized actions, while the **mean field model (an LLM) infers how these actions affect collective dynamics and outputs the key population signals (=mean field) that influences future individual decisions.** Unlike prior methods that manually encode interaction heterogeneity or assign fixed influence weights, our approach automatically generates these effects at each step.
>
> **2. Empirical**: **Experimental results in Appendix C and Fig. 8 (4) have already demonstrated disproportionate influence**: when the famous Chinese sports commentator "@Huang Jianxiang" was referenced to verify the truth, the population signal shifted toward skepticism, leading to reduced rumor spread.
>
> **3. We added two sets of controlled experiments** to demonstrate that the mean field model can capture heterogeneous interactions and disproportionate influence without any manual weighting or structural assumptions:
>
> **(1) Heterogeneous Interactions**
>
> We compare the effects of two different actions taken by the same **high-influence user**.
>
> - **State**: A non-verified user with high influence and many friends.
>
> - **Previous Mean Field**:
>
>   `Most users express opposition and criticism toward the event, regarding it as unfair competition.`
>
> - **Action A**: `"Repost"`
>
> - **Action B**: `"Looking forward to the truth!"`
>
>
>
> The mean field model takes the individual’s state and action as input and updates the mean field accordingly:
>
> - **After Action A**:
>
>   `"Most users are criticizing the event rather than seeking factual verification."`
>
> - **After Action B**:
>
>   `"Users call for an investigation to uncover the cause. … Many suspect that this might be a genuine news report."`
>
> Due to rebuttal length limits, only excerpts of updated mean field are shown here; the full version will appear in the revised paper.
>
> **(2) Disproportionate Influence**
>
>
>
> We compare the effects of the **same action** taken by two users: one with **high** influence and one with **low** influence.
>
>
>
> - **Action**: `"Looking forward to the truth!"`
> - **Previous Mean Field**: *(Same as in Case Study 1)*
>
>
>
> - **Agent States**:
>
> - - **High-Influence User**: A non-verified user with high influence and many friends.
>   - **Low-Influence User**: Same description, but with low influence and few friends.
>
>
>
> **Updated Mean Field:**
>
> - After **High-Influence User** acts: `"Many suspect that this might be a genuine news report."`
>
> - After **Low-Influence User** acts: `"A few users in the comments express a desire to uncover the truth."`
>
> ---
>
> ###
>
> > **Q3: "How do you address potential bias from using GPT-4o-mini for semantic evaluation? ..."**
>
> **Answer:** To balance **evaluation cost and speed**, we chose GPT-4o-mini for semantic evaluation. We validated its reliability through **human evaluation** on 100 samples by domain experts (social science scholars, junior faculty, and senior PhD students), comparing GPT-4o-mini’s outputs with human judgments. We also tested other LLMs (GPT-3.5-Turbo, GPT-4o, GPT-4). Summary:
>
> | Metric|GPT-4o-mini| GPT-4   | GPT-3.5-Turbo | GPT-4|
> | - | - | - | - | - |
> | Agreement w/ Human (%) | 91.25 | 89.38   | 79.13 | **94.13** |
> | Price ($/M tokens)| **0.15/0.60** | 2.5/10  | 0.5/1.5 | 30/60   |
> | Speed (s/sample)| 1.25 | 0.96    | **0.83** | 4.02|
>
> Since semantic evaluation is not a challenging task for today’s advanced LLMs, **GPT-4o-mini balances accuracy and cost effectively**, offering higher human alignment than GPT-3.5-Turbo at a much lower cost. With an optimized prompt (Appendix D.3), GPT-4o-mini provides the best cost-effectiveness, **especially considering its low price**, making it ideal for large-scale simulations. We will include full results in the revised paper.
>
> ---
>
>
>
> > **Limitation: "computational complexity analysis and testing on truly large populations."**
>
> **Answer:**
>
> **1. Computational Complexity Analysis**:
>
> MF-LLM aims to capture dynamic agent interactions, which are fundamental to the formation of collective behavior. **Modeling all agent–agent pairwise interactions requires O(N²) complexity, while MF-LLM reduces this to O(N) linear complexity.** Specifically, the policy model generates one personalized action per agent, leading to O(N) LLM inferences. The mean field model updates the population signal once per batch of K agents, resulting in O(N/K) mean field inferences.
>
> Overall, MF-LLM reduces total LLM inference complexity to O(N) while still modeling agent interactions and producing population-level dynamics. This **contrasts with baselines** that either ignore agent interactions entirely or approximate them through manual prompt engineering without dynamic modeling.
>
> **2. Testing on Large Populations**:
>
> In the original paper, the Weibo dataset naturally limited simulation horizons to ~900 steps due to the discussion length of real events. To further validate scalability, we extend evaluation to the English Twitter dataset with long-horizon discussions (up to 2,000 steps). **Results show that while error increases slightly with horizon, MF-LLM consistently outperforms baselines.**
>
> | Method | KL↓ | WD↓  | DTW↓ | NLL↓  | Macro F1↑  | Micro F1↑  |
> | - | - | - | - | - | - | - |
> | State| 1.1701 | 0.0714 | 0.1110  | 4.2050 | **0.3773** | 0.8456   |
> | Recent| 1.8701 | 0.0685 | 0.1178 | 4.1000  | 0.3210     | 0.8533 |
> | Popular | 1.4291 | 0.0722 | **0.0937** | 4.0863  | 0.3475     | 0.8438     |
> | **MF-LLM** | **1.1053** | **0.0685** | 0.0965 | **4.0236** | 0.3569     | **0.8537** |
>
> ---
>
> **We hope these clarifications and new results will encourage you to reconsider your score. If you have any further questions, we are happy to discuss them.**

---

> ### Author Response · Authors · 2025-08-04
> **Hope to Get Your Reply**
>
> **Dear Reviewer f92d,**
>
> Once again, we sincerely thank you for the time and effort you have dedicated to reviewing our paper. As the discussion period is approaching its close, we are eager to receive your feedback on our response. We fully understand your busy schedule, but we would greatly appreciate it if you could take our response into account when updating your rating and discussing with the AC and other reviewers.
>
> In addition to the experiments **already presented in the rebuttal**, including:
>
> - Experiments on English datasets (Twitter) and market behavior simulation,
> - Case studies of heterogeneous interactions and disproportionate influence,
> - Comparison of GPT‑4o‑mini for semantic evaluation with human experts, GPT‑4, GPT‑4o, and GPT‑3.5,
> - Simulation experiments extending to 2000 steps,
>
> we have **added further experiments** to strengthen our claims:
>
> ------
> > **Weakness & Q2: "How would you validate MF-LLM's performance on entirely different cultural contexts where social interaction norms differ significantly?"**
>
>  We **conducted additional evaluations** of MF‑LLM (without fine‑tuning) on the **Twitter dataset** using `GPT‑4o‑mini`, `GPT‑4o`, and `DeepSeek‑R1‑Distill‑Qwen‑32B` backbones. Results show that MF‑LLM consistently generalizes across cultural contexts.
>
> | Method | KL↓        | WD↓        | DTW↓       | Macro F1↑  | Micro F1↑  |
> | - | - | - | - | - | - |
> | `GPT‑4o‑mini`    |   |   |   |   |   |
> | State  | 6.8382     | **0.1469** | 0.4513     | 0.2808     | 0.6282     |
> | Recent | 6.5273     | 0.1491     | 0.4444     | 0.2843     | 0.6361     |
> | Popular| 7.5150     | 0.1634     | 0.4251     | 0.2526     | 0.6112     |
> | **MF‑LLM** | **5.9322** | 0.1506     | **0.3300** | **0.2934** | **0.6403** |
> | `GPT‑4o`      |   |   |   |   |   |
> | State  | 7.8105     | 0.1711     | 0.5178     | 0.2388     | 0.5962     |
> | Recent | 7.2690     | 0.1607     | 0.4576     | 0.2574     | 0.6200     |
> | Popular| 8.8492     | 0.1818     | 0.5198     | 0.2138     | 0.5798     |
> | **MF‑LLM** | **2.2669** | **0.1085** | **0.2211** | **0.4103** | **0.7249** |
> | `DeepSeek‑R1‑Distill‑Qwen‑32B` |   |   |   |   |   |
> | State  | 1.7496     | 0.1105     | 0.2847     | 0.4173     | 0.6999     |
> | Recent | 2.4299     | 0.1192     | 0.3462     | 0.3941     | 0.6829     |
> | Popular| 1.4311     | **0.0905** | 0.2198     | 0.4424     | **0.7256** |
> | **MF‑LLM** | **1.1814** | 0.0922     | **0.2042** | **0.4483** | 0.7204     |
>
> ------
> > **Limitation: "computational complexity analysis and testing on truly large populations."**
>
> **(1) Testing on Large Populations**
>
> We also **extended the simulation length** (Qwen2‑1.5B‑Instruct, without fine‑tuning) to the equivalent of **4000** agent updates (previously reported as “steps”). **Results** for 500–4000 agents are shown below, confirming stable performance over long horizons.
>
> | Agents | KL↓    | WD↓    | DTW↓   | NLL    | Macro F1↑ | Micro F1↑ |
> | - | - | - | - | - | - | - |
> | 500   | 1.2582 | 0.0645 | 0.1196 | 3.9916 | 0.3688    | 0.8644    |
> | 1000  | 1.4638 | 0.0729 | 0.1220 | 4.0112 | 0.3643    | 0.8436    |
> | 2000  | 1.1053 | 0.0685 | 0.0965 | 4.0236 | 0.3569    | 0.8537    |
> | 3000  | 1.0668 | 0.0650 | 0.0837 | 4.0142 | 0.3592    | 0.8611    |
> | 4000  | 1.5905 | 0.0698 | 0.1025 | 4.0237 | 0.3548    | 0.8376    |
>
> **(2) Runtime complexity**
>
> To complement the theoretical **O(N) complexity analysis** presented earlier, we **conducted empirical measurements** to verify runtime scaling in practice. We measured simulation runtime for varying numbers of agent updates on `Qwen2‑1.5B‑Instruct` (Twitter dataset, without fine‑tuning). Experiments were run on a single NVIDIA A100 GPU (40 GB memory).
>
> | Agents | Time (s) | Time / Agent (s) |
> |- | - | - |
> | 512    | 280.88   | 0.55 |
> | 1008   | 561.89   | 0.56   |
> | 1504   | 832.31   | 0.55|
> | 2000   | 1113.41  | 0.56|
> | 2512   | 1404.34  | 0.56|
> | 3008   | 1687.50  | 0.56|
> | 3504   | 1966.29  | 0.56|
> | 4000   |2249.15  | 0.56|
>
> **Finding:** Runtime increases **linearly** with the number of agents, maintaining a nearly constant time per agent (~0.56 s), which primarily reflects the cost of repeated LLM calls. This empirical result validates the theoretical **O(N)** complexity, providing experimental evidence for MF‑LLM’s scalability.
>
> ------
>
> **We hope that these efforts will alleviate your concerns regarding MF‑LLM. Your feedback is highly valuable to us, and we would appreciate any updates or further guidance you might have regarding our revisions and responses.**

---

> ### Author Response · Authors · 2025-08-07
> **Hope Our Rebuttal Addresses Your Concerns – Looking Forward to Your Feedback**
>
> **Dear reviewer f92d**,
>
> As the reviewer–author discussion deadline is approaching, we will be online waiting for your feedback on our rebuttal, which we believe has fully addressed your concerns.
>
> We would highly appreciate it if you could take into account our response when updating the rating and having discussions with AC and other reviewers.
>
> Thank you so much for your time and efforts. Sorry for our repetitive messages, but we're eager to ensure everything is addressed.
>
> Authors of Paper #4199

---

> ### Author Response · Authors · 2025-08-08
> **Summary of Our Updates and Responses – Awaiting Your Feedback**
>
> **Dear Reviewer f92d,**
>
> We sincerely thank you for your valuable comments on improving our work. We would like to respectfully highlight the updates made in response to your suggestions and to address your concerns:
>
> 1. **Added evaluations on diverse datasets** (English Twitter, market simulation) using `Qwen2-1.5B-Instruct`, `GPT-4o-mini`, `GPT-4o`, and `DeepSeek-R1-32B`, validating MF-LLM’s performance across entirely different cultural contexts.
> 2. **Clarified the mechanism with mean-field approximation**,and **provided empirical evidence** (Appendix C, Fig. 8) and **two sets of controlled experiments**, showing MF-LLM’s ability to handle heterogeneous interactions and disproportionate influence.
> 3. **Added comparison** of `GPT-4o-mini` with **human experts**, `GPT-4`, `GPT-4o`, and `GPT-3.5`, confirming it offers the best trade-off between accuracy, cost, and speed for semantic evaluation.
> 4. **Added computational complexity analysis** (O(N²) → O(N)) and **empirical runtime measurements**, verifying linear scaling in practice.
> 5. **Extended to larger populations** (up to 4000 agents) and showed that the error increases only marginally from 500 to 4000 agents.
>
> We respectfully hope that you will take these updates and our responses into account when making the final decision. As the author–reviewer discussion period is coming to an end, we look forward to further engaging with you. We would be happy to continue the discussion and provide additional information if you have any further concerns or questions.
>
> Authors of Paper #4199

---

### Decision · Program_Chairs · 2025-09-17

**Decision:**

Accept (poster)

**Comment:**

The authors present a new method for LLM-based social simulation for modeling phenomena like rumor spread and decision making in social networks. The contributions include introducing a “mean-field” LLM agent to summarize agent observables to align with population data, and an objective based the information-bottleneck to train this agent. Empirical results show improvements vs. existing LLM social simulation frameworks.

All reviewers appreciated the overall approach, describing it as principled, interesting, and novel. They also found the experiments on the Weibo dataset to be convincing, concluding that the model showed robust performance and improvements over baselines. Two reviewers commented that the evaluation on a single dataset was limited, and encouraged authors to explore one or more settings beyond Weibo. During the rebuttal period, the authors presented results for two additional settings including English language tweets. Other questions about strength of evidence such as computational complexity were also addressed during the rebuttal. One reviewer commented that clarity could be improved, especially giving more direct explanations of the method and agent interactions. Clarity was also reflected in several of the questions asked by reviewers. The author responses in the rebuttal were generally adequate on these issues, and the authors are encouraged to improve these explanations in the final paper.

Overall, reviewers were satisfied with the rebuttal and two raised their scores, leading to a positive overall assessment.